# Comparing soil carbon loss through respiration and leaching under extreme precipitation events in arid and semi-arid grasslands

Ting Liu[1#], Liang Wang[1#], Xiaojuan Feng[1,2*], Jinbo Zhang[3], Tian Ma[1,2], Xin Wang[1,2], Zongguang Liu[1]

[1] State Key Laboratory of Vegetation and Environmental Change, Institute of Botany, Chinese Academy of Sciences, Beijing 100093, China,
[2] University of Chinese Academy of Sciences, Beijing, China,
[3] School of Geography Sciences, Nanjing Normal University, Nanjing 210023, China

# co-first authors with equal contributions

* *Correspondence to*: Xiaojuan Feng (xfeng@ibcas.ac.cn)

**Abstract.**

Respiration and leaching are two main processes responsible for soil carbon loss. While the former has received considerable research attention, studies examining leaching processes are limited especially in semiarid grasslands due to low precipitation. Climate change may increase the extreme precipitation event (EPE) frequency in arid and semiarid regions, potentially enhancing soil carbon loss through leaching and respiration. Here we incubated soil columns of three typical grassland soils from Inner Mongolia and Qinghai-Tibetan Plateau and examined the effect of simulated EPEs on soil carbon loss through respiration and leaching. EPEs induced transient increase of $CO_2$ release through soil respiration, equivalent to 32% and 72% of the net ecosystem productivity (NEP) in the temperate grasslands (Xilinhot and Keqi) and 7% of NEP in the alpine grasslands (Gangcha). By comparison, leaching loss of soil carbon accounted for 290%, 120% and 15% of NEP at the corresponding sites, respectively, with dissolved inorganic carbon (DIC, biogenic DIC + lithogenic DIC) as the main form of carbon loss in the alkaline soils. Moreover, DIC loss increased with re-occuring EPEs in the soil with the highest pH due to elevated contribution of dissolved $CO_2$ from organic carbon degradation (indicated by DIC-$\delta^{13}$C). These results highlight that leaching loss of soil carbon (particularly in the form of DIC) is important in the regional carbon budget of arid and semiarid grasslands, and also imply that SOC mineralization in alkaline soils might be under-estimated if only measured as $CO_2$ emission from soils into the atmosphere. With a projected increase of EPEs under climate change, soil carbon leaching processes and its influencing factors warrant better understanding and should be incorporated into soil carbon models when estimating carbon balance in grassland ecosystems.

# 1 Introduction

Soils store approximately 2500 Pg of carbon (including organic and inorganic carbon) globally, equivalent to 3.3 and 4.5 times the carbon in the atmosphere (760 Pg) and terrestrial plants (560 Pg), respectively (Lal, 2004). Slight variations of the soil carbon pool will hence severely influence atmospheric $CO_2$ concentrations and have important implications for climate change
(Davidson and Janssens, 2006; Trumbore and Czimczik, 2008). Respiration and leaching are two main processes responsible for soil carbon loss. While respiration has received considerable research attention (Raich and Schlesinger, 1992; Raich and Potter, 1995; Hoover et al., 2016; Burri et al., 2015; Escolar et al., 2015), leaching is relatively poorly constrained despite its importance in certain ecosystems (cr et al., 2007; Battin et al., 2008; Liu et al., 2017). For instance, soil carbon leached from forests, grasslands, and croplands is estimated to be 15.1, 32.4, and 20.5 g C $m^{-2}$ $yr^{-1}$ across Europe, representing 4%, 14%,
and 8% of net ecosystem exchange (NEE), respectively (Kindler et al., 2011). Additionally, leaching of carbon previously preserved in surface litter and soil layers is believed to be a main source of dissolved organic and inorganic matter in inland waters (Spencer et al., 2008). In particular, soil inorganic carbon (SIC) that occurs widely in the arid and semiarid regions is more prone to leaching than organic carbon during sporadic high precipitation events (Lal and Kimble, 2000). Despite the importance of leaching loss in regional soil carbon budget, very few detailed data exist to investigate and compare the relative
contribution of respiration and leaching processes to soil carbon loss.

Climate change is reported to increase the frequency as well as intensity of extreme precipitation events (EPEs; Knapp et al., 2002; Goswami et al., 2006; Parry et al., 2007; Min et al., 2011; Reichstein et al., 2013), especially in arid regions (Donat et al., 2017). In northwestern China, the frequency and intensity of EPEs have showed an increasing trend in the recent 50 years, constituting a much higher proportion of total precipitation than light precipitation events (Liu et al., 2005; Chen et al., 2012;
Wang et al., 2012; Fu et al., 2013; Wang et al., 2014). Increasing EPEs will not only enhance soil carbon leaching but also affect soil respiration processes through increasing soluble substrates for microbial decomposition and potentially inducing hypoxic conditions (Knapp et al., 2002; Harper et al., 2005; Morel et al., 2009; Unger et al., 2010). Hence, it is critical to evaluate the effects of EPEs on soil respiration and leaching processes in order to better understand the impact of climate change on terrestrial carbon cycling, especially in the arid and semiarid regions.
Grasslands, containing 20% of global soil carbon pool, are the most widespread ecosystems in arid and semiarid regions globally (Jobbagy and Jackson, 2000). The deposition rate of carbonate is relatively high in the grassland soils with a high alkalinity and aridity (Lal, 2008; Yang et al., 2012), and hence SIC is the major form of soil carbon in many grasslands (Mi et al., 2008). SIC storage in China is approximately 53.3−77.9 Pg (Li et al., 2007; Mi et al., 2008), 54% of which is mainly distributed in the temperate and alpine grasslands located in Inner Mongolia and Qinghai-Tibetan Plateau (Mi et al., 2008).
From 1980s to 2000s, SIC in the topsoil of Chinese grasslands was estimated to decrease by 26.8 g C $m^{-2}$ $yr^{-1}$, mainly attributed to soil acidification (Yang et al., 2012). Alternatively, precipitation is one of the main factors influencing the distribution and storage of SIC in arid and semiarid regions (Batjes, 1998; Lal and Kimble, 2000). Mi et al. (2008) found that 84% of SIC in

China was distributed in areas with a mean annual precipitation (MAP) of $< 500$ mm and that SIC content decreased significantly with the increase of MAP. Given the high leaching potential of SIC in grassland soils under altered precipitation patterns in the future, we hypothesize that EPEs may significantly enhance SIC loss through leaching processes and further reduce SIC storage in grasslands.

In this study, soils were collected from varied depths of three typical temperate and alpine grasslands in Inner Mongolia and Qinghai-Tibetan Plateau to construct soil columns for a laboratory incubation study. Using simulated EPEs, we examined soil carbon loss through respiration and leaching processes and compared their fluxes after EPEs. In addition, leaf litter of a C4 grass was added to the surface of one set of soil columns to compare soil carbon loss from bare versus litter-covered soils and to estimate the contribution of litter-derived carbon to soil respiration after EPEs. Our research objectives were: (1) to

investigate the influence of EPEs on soil respiration; (2) to quantify the loss of SIC and soil organic carbon (SOC) through leaching during EPEs; and (3) to compare the relative importance of respiration and leaching in EPE-induced soil carbon loss from grassland soils.

## 2 Materials and Methods

### 2.1 Study area

For the incubation experiment, soils were collected from three different sites of temperate and alpine grasslands of China with varied environmental characteristics. Temperate grasslands were sampled near Xilinhot (XLHT, 116˚22' E, 44˚8' N, mean elevation of 1170 m) and Keqi (KQ, 117˚15' E, 43˚18' N, mean elevation of 1250 m) within the arid and semiarid regions of Inner Mongolia (Fig. S1) with MAP of 299 and 402 mm and mean annual temperature (MAT) of 1.2 and 0.4˚C, respectively. Soil in this region is mainly chestnut soil, classified as Calcic Chernozems according to the World Reference Base for Soil

Resources (Steffens et al., 2008; IUSS working group WRB, 2015), with *Stipa klemenzii, Stipa Goboca, Stipa breviflora,* and *Stipa glareosa* as the dominating species (Sui and Zhou, 2013). The alpine grassland was sampled in Gangcha (GC, 100˚7' E, 37˚19' N, mean elevation of 3500 m) located north of the Qinghai Lake on the northeastern edge of the Qinghai-Tibetan Plateau. The GC site has an MAT of 0.4˚C, an MAP of 370 mm and a mean annual evaporation (MAE) of 607 mm. Soils at this site are mainly Gelic Cambisol (IUSS working group WRB, 2015), with *Potentilla ansrina Rosaceae, Elymus nutans*

*Griseb,* and *Deyeuxia arundinacea* as the dominant species.
Soils were collected by digging soil pits of 25 cm $\times$ 25 cm $\times$ 70 cm from the temperate (XLHT and KQ) and alpine (GC) sites in October, 2014 and August, 2015, respectively. At each site, three plots (200 m $\times$ 200 m) were selected ($> 200$ m in between) with three random soil pits (distance of ~ 5 m in between) sampled within each plot. Soils from the same depth (0–20, 20–40, and 40–60 cm) of the three soil pits were mixed *in situ* for each plot, shipped back to the laboratory immediately, and stored

at 4˚C before the experiment started within one month. As a result, each sampling site had three "true" replicates from the field for the soil column experiment.

**2.2 Soil column experiment and simulated EPEs**

For the laboratory experiment, we reconstructed soil columns of similar structures and texture under controlled conditions and used gravity to collect soil leachates. This approach is commonly used in process-related research (Hendry et al., 2001; Thaysen et al., 2014; Artiola and Walworth, 2009; Aslam et al., 2015) as it minimizes experimental errors and bias caused by unknown factors including soil heterogeneity and microbial community variations. It is also more favourable in terms of quantifying soil carbon leaching loss as it circumvents pore-water contamination by vacuum suction in the field. In particular, leachate sampling by gravity from soil columns prevents alterations to DIC concentrations, which may be caused by $CO_2$ outgassing using vacuum suction in field studies. Artificial soil columns were constructed in the laboratory with polymethyl methacrylate frames (diameter: 10 cm; height: 70 cm; Fig. 1). The bottom of each column had an aperture (inner diameter: 0.6 cm; height: 3 cm) for the collection of soil leachates, and the column top was fitted with an airtight lid connected to two tubes for gas exchange and collection. Empty columns were soaked in 0.1 mM hydrochloric acid (HCl) solutions for 12 h and rinsed with distilled water before use. Column bottoms were packed with pre-cleaned quartz sand (5-cm thick; soaked in 0.1 mM HCl and combusted at 450˚C for 6 h before use) with a layer of nylon net (pore size: 150 μm; diameter: 10 cm) on both sides to prevent the movement of soil particles. Subsequently, soils were passed through 2-mm sieves with roots removed and packed into each column at the corresponding depths (in the sequence of 40−60, 20−40, and 0−20 cm). Soils were compacted gently to maintain a similar bulk density as in the field (Table 1). Water content of each soil layer was separately adjusted to 60% of the maximum water holding capacity (Table 1) to provide an ideal moisture condition for microbial growth (Howard and Howard, 1993; Rey et al., 2005). There was a 10-cm headspace unfilled with soil for each column.

Six soil columns (one litter-amended and one non-amended column for each of the three sampling plots) were set up for each site as described above, and pre-incubated for two weeks in the laboratory to allow the recovery of microbial communities after disturbance. Subsequently, leaf litter of a C4 grass, *Cleistogenes squarrosa*, a dominant species in the grasslands of northern China (Tian et al., 2015), was added to the surface of three columns in an amount equivalent to the aboveground biomass in the field (1.26 g for the XLHT and KQ sites and 1.59 g for the GC site; Bai et al., 2008). The isotopic signal of the leaf litter ($δ^{13}C$ of −16.2‰) would allow us to estimate the contribution of litter-derived $CO_2$ to total soil respiration. The columns were pre-incubated again for seven days. Basal respiration rate was measured by collecting $CO_2$ gas in the column headspace after 4 h of incubation. Temperature was recorded every day during the whole incubation period (23 ± 1˚C).

According to historical precipitation records (Fig. S2), more than 70% of the annual precipitation occurs from June to August in the study area, mainly in the form of 2-4 heavy precipitation events. Therefore, a total of three EPEs were simulated over a period of 2 months for each soil using artificial rainwater prepared according to rainwater's composition at the corresponding sites (pH of 7.3; Table S1; Tang et al., 2014; Zhang et al., 2013). A maximum rainfall intensity of ~100 mm per precipitation event has been recorded in the past two decades in the study area (Fig. S2) and is predicted to increase by 18.1% in the late 21st century in north China (Chen et al., 2012). Hence, approximately 1 L of rainwater (rainfall of ~127 mm), comparable to 30% of the MAP of the investigated sites, was added to the surface of each soil column over 3–4 h at rates of one drop per

second using syringes and allowed to leach through the column to be collected with a clean beaker within 12–14 h. The leachates were weighed, filtered through a 0.45-μm PTFE syringe filter and analyzed for dissolved organic carbon (DOC) and dissolved inorganic carbon (DIC) concentrations immediately. To monitor soil respiration every 1–2 days following each EPE, soil columns were first aerated for 1 h using $CO_2$-depleted air that had been passed through saturated sodium hydroxide (NaOH)

solutions (twice; Fig. 1) and then incubated for 4 h with lids closed. $CO_2$ gas in the column headspace was collected by gas-tight syringes for the subsequent measurement. After collection of $CO_2$ gas, the lids were open to allow the exchange with the ambient air. Soil respiration was monitored for 30 days after the first EPE and observed to stabilize approximately on the 20th day (Fig. S3). Hence, the first, second, and third EPEs were conducted on the 1st, 31st, and 51st day of incubation, and the $CO_2$ measurement was conducted for approximately 30, 20, and 20 days after the first, second, and third EPEs, respectively.

Basal respiration was considered to be represented by the stabilized respiration rate at the end of each EPE cycle. In addition, due to constrained time and logistic reasons, the soil respiration after the second EPE in the KQ soils was not monitored, and the cumulative respiration after the second EPE was calculated as the average respiration after the first and third EPEs in the KQ soils.

## 2.3 Sample analyses

Soil pH was measured at a soil:water ratio of 1:2.5 (w:v) using a pH meter (Sartorius PB−10). Soil texture was examined by laser diffraction using Malvern Mastersizer 2000 (Malvern Instruments Ltd., UK) after removal of organic matter and calcium carbonates. Soil field water content was determined by difference between moist and dried soils (dried at 105˚C for 8 h). Maximum water holding capacity was estimated by weighing soils before and after removal of redundant water from fully soaked soils (in water for 8 h). For SOC analysis, dried soils were decarbonated by exposure to concentrated HCl vapor for 72

20 h, followed by saturated NaOH solutions for 48 h to neutralize extra HCl, and then dried at 45°C. Total soil carbon, SOC (after decarbonation) and nitrogen (N) contents were measured by combustion using an elemental analyser (Vario EL III, Elementar, Hanau, Germany). SIC was calculated as the difference between total carbon and SOC contents. Small aliquots of the soil leachates were analyzed immediately on a Multi N/C 3100-TOC/TN Analyzer (Analytik Jena, Germany) for DIC and DOC concentrations (with the latter acidified to pH < 2 with concentrated HCl before analysis). It should be mentioned that the DIC

concentration may vary due to exchanges between dissolved and atmospheric $CO_2$ during leachate collection. However, potential contribution from this process was < 7% owing to the low proportions of dissolved $CO_2$ in total DIC of our samples (Table S2) as calculated according to Ran et al. (2015). $CO_2$ concentration in the soil column headspace was determined by gas chromatograph (Agilent 7890A, USA) coupled with a flame ionization detector (FID).

To examine the contribution of SOC- and litter-derived carbon to soil respiration, the $\delta^{13}C$ values of SOC and $CO_2$ gas were

30 determined on an isotope ratio mass spectrometer (Delta plus xp, Thermo, Germany) with a precision of ± 0.2‰. To estimate the contribution of SOC degradation to leached DIC, the $\delta^{13}C$ values of DIC were determined on a Picarro isotopic $CO_2$ analyzer equipped with an automated DIC sample preparation system (AutoMate) based on wavelength scanned cavity ring down spectroscopy technique (Picarro AM-CRDS, USA). The precision for the DIC-$\delta^{13}C$ measurement was ± 0.3‰. Due to

budget constraints and logistic reasons, we only measured the $\delta^{13}C$ of the respired $CO_2$ in the GC soils during the first EPE and the leached DIC in the XLHT soils.

## 2.4 Data analysis and statistics

The relative contribution of litter- and SOC-derived $CO_2$ to total respired $CO_2$ in the litter-amended soils was estimated using the following mass balance model:

$$f_{\text{litter-derived}} + f_{\text{SOC-derived}} = 1 \tag{1}$$

$$f_{\text{litter-derived}} \times \delta^{13}C_{\text{litter-derived}} + f_{\text{SOC-derived}} \times \delta^{13}C_{\text{SOC-derived}} = \delta^{13}C_{\text{respired-CO2}} \tag{2}$$

where $f_{\text{litter-derived}}$ and $f_{\text{SOC-derived}}$ are the proportion of litter- and SOC-derived $CO_2$ in the total respired $CO_2$; $\delta^{13}C_{\text{litter-derived}}$ is the $\delta^{13}C$ value of litter-derived $CO_2$, equivalent to $-16.25$‰; $\delta^{13}C_{\text{SOC-derived}}$ is the $\delta^{13}C$ value of SOC-derived $CO_2$, which assumes the same value as that in the non-amended soils at the beginning of incubation ($-23.1$‰) according to Cerling et al. (1991); $\delta^{13}C_{\text{respired-CO2}}$ is the measured $\delta^{13}C$ of respired $CO_2$.

Similarly, the relative contribution of lithogenic carbonate and biogenic DIC derived from SOC degradation to leached DIC was assessed according to the following isotopic mass balance model:

$$f_{\text{carbonate}} + f_{\text{biogenic-DIC}} = 1 \tag{3}$$

$$f_{\text{carbonate}} \times \delta^{13}C_{\text{carbonate}} + f_{\text{biogenic-DIC}} \times \delta^{13}C_{\text{biogenic-DIC}} = \delta^{13}C_{\text{DIC}} \tag{4}$$

where $f_{\text{carbonate}}$ and $f_{\text{biogenic-DIC}}$ are proportion of carbonate- and biogenic DIC in total DIC; $\delta^{13}C_{\text{carbonate}}$ is the $\delta^{13}C$ value of soil carbonate, equivalent to $0$‰ (Edwards and Saltzman, 2016); and $\delta^{13}C_{\text{biogenic-DIC}}$ is the $\delta^{13}C$ value of biogenic carbonate/bicarbonate derived from the dissolution of $CO_2$ produced by SOC degradation, which is estimated to shift by approximately 8‰ compared with the $\delta^{13}C$ value of soil-respired $CO_2$ ($-24$‰ here) due to isotope fractionation during $CO_2$ dissolution (Zhang et al., 1995). Hence, $\delta^{13}C_{\text{biogenic-DIC}}$ is estimated to be $-16$‰. $\delta^{13}C_{\text{DIC}}$ is the measured $\delta^{13}C$ signature of leached DIC. Isotopic fractionation of leached DIC due to $CO_2$ loss in an open system is insignificant when the partial pressure of $CO_2$ ($pCO_2$) in the solution is lower than twice that of the surrounding atmosphere (Hendy, 1971; and Doctor et al., 2008). In the present study, $pCO_2$ in the XLHT leachates was low ($\sim 400$ μatm assuming alkalinity equals to DIC concentration; Table S2) due to its high pH, low soil respiration and dilution of dissolved $CO_2$ under EPE. Thus, we considered the influence of $CO_2$ outgassing on the $\delta^{13}C$ of leached DIC to be negligible.

EPE-induced $CO_2$ release via respiration was assessed following two steps. First, cumulative respiration during the first 20 days after each EPE (until respiration rate stabilized) was calculated. Second, difference between the measured cumulative respiration and that estimated using the stabilized basal respiration rate after each EPE was calculated as the EPE-induced $CO_2$ release.

Independent samples T test (group size = 2) and One-way ANOVA analysis (group size > 2) was used to compare the dissolved carbon concentrations and fluxes among different columns. Linear regression analysis was used to assess correlations between

leachate carbon flux and influencing factors (carbon content, soil pH, soil texture, etc.). All these analyses were performed using IBM SPSS Statistics 22. Differences and correlation s are considered to be significant at a level of $p < 0.05$.

## 3 Results and Discussion

### 3.1 Bulk properties of grassland soil samples

In the investigated grassland soils, SOC represented 59−99% of soil carbon and exhibited $\delta^{13}C$ values typical of C3 plant inputs (ranging from −24.1‰ to −26.3‰; Table 1). The XLHT soil had a much lower SOC and nitrogen (N) contents than the KQ and GC soils despite a similar soil texture ($p < 0.05$; Table 1). The SOC:N ratio was also lowest in XLHT (7.09−8.03), indicating a more decomposed state of soil organic matter (Weiss et al., 2016). Conversely, the SIC content was highest in XLHT and lowest in KQ, in line with soil pH variations at these sites, i.e., lowest pH in KQ and highest in XLHT. This correlation of SIC with soil pH is consistent with the results of Shi et al. (2012), showing that pH is the most important factor controlling SIC variation across the Mongolian and Tibetan grasslands. In terms of depth variations, soils became coarser with depth in XLHT and GC, but became finer with increasing depth in KQ. The SOC and N contents decreased with depth in all soils due to declining plant inputs ($p < 0.05$; Table 1), while the SOC:N ratio remained relatively similar (except a small decrease with depth in XLHT). By contrast, XLHT and GC soils showed an increasing SIC content with depth ($p < 0.05$; Table 1), because SIC, with a good solubility, is prone to leaching from the topsoil and subsequently precipitates in the deeper soil (Mi et al., 2008; Tan et al., 2014). The KQ soil, showing an almost neutral pH, had an invariant SIC content and pH with depths. Overall, the varied properties (including SOC, SIC, pH, etc.) of these soils allowed us to compare the effects of EPEs on soil respiration and leaching processes in different grassland soils.

### 3.2 EPE-induced changes to soil respiration

Shortly after each simulated EPE, soil respiration was similar to or lower than basal respiration (Fig. S3). The latter case may be attributed to hypoxic conditions induced by water saturation during EPEs (Hartnett and Devol, 2003; Jessen et al., 2017). Subsequently, soil respiration increased and peaked after approximately one week due to the recovery of microbial activity with improved soil aeration (Borken and Matzner, 2009). It then decreased to a constant level approximately 20 days after each EPE (Fig. S3). The transient increase of respiration was consistent with the "Birch Effect" (Birch, 1964), i.e., a pulse of soil respiration after rewetting events due to resuscitation of microorganisms and improved diffusive transport of substrate and extracellular enzymes (Borken and Matzner, 2009; Navarro-García et al., 2012; Placella et al., 2012). The maximum soil respiration rates were 40.6 and 37.3 mg C m$^{-2}$ h$^{-1}$ after EPEs in the non-amended KQ and GC soils, respectively. These rates were significantly higher than that in the XLHT soil (13.7 mg C m$^{-2}$ h$^{-1}$), likely related to the higher SOC content in the former soils. The maximum specific soil respiration rates normalized to SOC were 2.2, 2.6, and 2.0 µg C g$^{-1}$ SOC h$^{-1}$ in the non-

amended GC, KQ, and XLHT soils, respectively. Therefore, SOC degradability was quite similar in the alpine and temperate grassland soils.

Total respired $CO_2$ was higher in the litter-amended than non-amended soils before and after EPEs (Fig. S6). The cumulative respired $CO_2$ in the litter-amended XLHT, KQ, and GC soils were 16.7, 54.8, and 44.6 g C m$^{-2}$ during three EPEs, 20%, 22%, and 15% higher than that of the non-amended soils, respectively. Due to the wide presence of litter coverage in our studied soils, litter effect on soil respiration should be considered when estimating carbon budgets for these grassland soils. The higher total respired $CO_2$ in litter-amended soils might be caused by one or two following reasons: (1) the degradation of labile components in the fresh litter; (2) induced priming effects due to the addition of an easily available energy source (Fröberg et al., 2005; Ahmad et al., 2013). To distinguish the influences of above two reasons on total respired $CO_2$ and further differentiate the contribution of litter (C4) and SOC (C3) to the respired $CO_2$, we examined the $\delta^{13}C$ values of $CO_2$ evolved from the GC soils after the first EPE. On the first day after EPE, $CO_2$ from the non-amended and litter-amended GC soils had a $\delta^{13}C$ value of –23.1‰ and –18.7‰, respectively. The latter was close to the $\delta^{13}C$ signature of the added litter (–16.25‰). Using the two-endmember mixing model of Eq. (1) and (2), we calculated that litter contributed ~64% of the respired $CO_2$ in the litter-amended GC soils. However, along with the consumption of labile carbon in litter, the $\delta^{13}C$ signature of $CO_2$ decreased from –18.7‰ on Day 1 to –21.8‰ on Day 25 after EPE in the litter-amended soils (Fig. 2). Accordingly, the proportion of litter-derived $CO_2$ decreased from 64% to 20%. The litter-derived $CO_2$ flux in litter-amended GC soils was estimated to range from 7.0 to 17.5 mg C m$^{-2}$ h$^{-1}$, while the SOC-derived $CO_2$ flux increased from 6.2 to 15.7 mg C m$^{-2}$ h$^{-1}$ after the first EPE (Fig. S4). Compared with the SOC-derived $CO_2$ flux in non-amended GC soils (ranging from 17.2 to 27.1 mg C m$^{-2}$ h$^{-1}$), litter addition had a negative priming effect on the degradation of native SOC while increasing total respiration through labile litter degradation.

Using data shown in Fig. S3 and S5, we calculated that total EPE-induced $CO_2$ release during three EPEs was higher in the KQ and GC soils than in the XLHT soil ($p < 0.05$; Fig. 3a) with a lower SOC content and a lower SOC:N ratio (Table 1). However, the specific EPE-induced $CO_2$ release normalized to SOC content showed no significant difference in the non-amended soils among three sites (Fig. 3b), indicating that a similar proportion of SOC (~4%) was subject to EPE-induced $CO_2$ release in the alpine and temperate grassland soils (Fig. 3b). The total EPE-induced $CO_2$ release was significantly higher in the litter-amended KQ soils than the non-amended ones. Besides the availability of labile OC provided by litter, the higher total EPE-induced $CO_2$ in litter-amended KQ soils might be related to its relatively lower soil pH (~7.7) that facilitates the release rather than the dissolution of respired $CO_2$ (from both SOC and litter mineralization) in soil solution. In addition, KQ has the highest mean sand content (46.9%) among the three soils (Table 1), i.e., the least possible mineral protection on labile OC dissolved from the litter, and this benefits the transport and mineralization of labile OC which meanwhile might play positive priming effects on the SOC mineralization. We therefore conclude that the KQ soil, with a coarser texture and a lower pH (Table 1), may have provided less sorptive protection for the labile DOC components after EPEs (Kell et al., 1994; Nelson et al., 1994) and allowed less dissolution of the respired $CO_2$, and hence showed a more responsive respiration to the precipitation

events. Consequently, we deduced that the availability of labile organic carbon, soil texture and pH are important factors influencing the total EPE-induced $CO_2$ release in temperate and alpine grassland soils.

## 3.3 EPE-induced leaching of soil carbon

During the first EPE, a total of 0.57, 0.56 and 0.73 L of leachates were collected from the XLHT, KQ, and GC soils, respectively. The leachate increased to 0.71, 0.94 and 0.87 L during the second EPE and was 0.69, 0.83 and 0.89 L during the third EPE, respectively (Fig. 4). Soil water content was set to be ~60% of max WHC before the first EPE, and leaching did not occur until soil water reached saturation. Therefore, the leachate volume was lowest during the first EPE and similar for the second and third EPEs. There were some variations in the volume of leachates from different soils, possibly related to preferential flows created during EPEs in the soil columns (McGrath et al., 2009) and water evaporation between EPEs. DIC was the main form of carbon in the leachates from the alkaline soils with a high SIC content (XLHT and GC) but low from the KQ soil with a neutral pH and low SIC content (Fig. 4). The resulting DIC flux was much higher for the XLHT soils (~21.3 g C m$^{-2}$) than the other two (2.9 g C m$^{-2}$ for KQ and 7.4 g C m$^{-2}$ for GC soils) during three EPEs, equivalent to five times of its DOC flux (3.8−4.2 g C m$^{-2}$, Fig. 4). In contrast, DIC flux in the KQ soils was only one third of its DOC flux during EPEs. The form of leached carbon was mainly linked to the amount of SOC and SIC in the columns (shown in Fig. S5).

Litter amendment did not increase DOC fluxes in any of the investigated soils but increased DIC fluxes leached from the KQ soil during the second and third EPEs and from the GC soil during the second EPE ($p < 0.05$, Fig. 4b-c). We postulate that, while litter contribution to DOC was minor, $CO_2$ derived from litter degradation contributed to dissolved $CO_2$ in soils and hence increased DIC in the leachates (Monger et al., 2015). This effect was not evident during the first EPE when litter decomposition just started and was not significant for the third EPE in the GC soil due to a high sample variability associated with the litter-amended soil (Fig. 4c). Due to the high SIC content in the XLHT soils (38.15 g per column) and the low litter-OC amendment (0.7 g per column), there was no significant difference of DIC fluxes between the non-amended and litter-amended XLHT soils (Fig. 4a). However, for the KQ soil having a relatively low SIC content similar to the added litter-OC (0.7 g per column; Table 1), litter amendment had a significant effect on the DIC flux (p < 0.05), increasing by 21 ± 13% and 15 ± 7% relative to the non-amended KQ soils during the second and third EPEs, respectively. There was also a 30 ± 19% increase in the DIC flux from the litter-amended GC soils relative to its non-amended counterpart during the second EPE. Therefore, litter amendment had a significant influence on DIC fluxes from soils with a relatively low SIC content (KQ and GC) under EPEs compared with the high-SIC XLHT soil.

Between different EPEs, leachate DOC fluxes did not vary in any of the investigated soils. By comparison, DIC fluxes increased in the XLHT soil from 4.5 g C m$^{-2}$ after the first EPE to 9.0 g C m$^{-2}$ after the third EPE ($p < 0.01$, Fig. 4). This increase may be caused by (i) an increased contribution of SOC degradation to soil DIC and/or (ii) an elevated dissolution of soil carbonates induced by higher soil $CO_2$ concentrations with repeated EPEs (Gulley et al., 2014; Ren et al., 2015). To evaluate these contributions, the $\delta^{13}C$ values of DIC were measured for the non-amended XLHT soil. The $\delta^{13}C$ of leached DIC ranged from −10.0‰ to −6.6‰ during the first EPE. Based on the isotopic mass balance of Eq. (3) and (4), lithogenic carbonate

(with a $\delta^{13}C$ value of 0‰) contributed 51% to the leached DIC while biogenic DIC produced by SOC degradation contributed 48% (Fig. 5). The $\delta^{13}C$ value of leached DIC decreased to −12.3‰ and −13.5‰ during the second and third EPEs, corresponding to a contribution of 77% and 84% by biogenic sources in the total DIC, respectively (Fig. 5). These results confirm our previous hypothesis that SOC decomposition contributed significantly to soil DIC fluxes. Combined with the total flux rate, we calculated that both lithogenic and biogenic DIC fluxes were ~2.1 g C m$^{-2}$ in the first EPE. Subsequently, lithogenic DIC flux decreased to ~1.3 g C m$^{-2}$ while biogenic DIC flux increased to 7.6 g C m$^{-2}$ in the third EPE. This demonstrates that the increased DIC flux with repeated EPEs was mainly derived from increased contribution of SOC mineralization. Interestingly, increasing DIC fluxes with repeated EPEs were not observed in the KQ and GC soils (Fig. 4) despite their higher SOC contents (Table 1) and $CO_2$ release rates (Fig. S3). Given that the XLHT soil had the highest soil pH, the high alkalinity may have favored the retention of respired $CO_2$ in the soil solution compared with the other soils (Parsons et al., 2004; Yates et al., 2013; Liu et al., 2015), leading to its high contribution to DIC fluxes.

Regardless of its source, the EPE-induced leaching loss of inorganic carbon was 31.5 and 10.6 μg DIC g$^{-1}$ soil from the alkaline XLHT and GC soils, respectively, approximately three and five times higher than the corresponding DOC leaching loss (5.9 and 3.9 μg DOC g$^{-1}$ soil, respectively). However, the KQ soil had a relatively lower EPE-induced DIC loss (4.4 μg DIC g$^{-1}$ soil) than the DOC leaching loss (11.6 μg DOC g$^{-1}$ soil) mainly due to its lower initial SIC content and relatively neutral soil pH value. Hence, total DIC (biogenic DIC + lithogenic DIC) was the main form of soil carbon loss in alkaline soils during EPEs. When the source of the leached DIC is taken into account, dissolution of $CO_2$ produced by SOC mineralization (biogenic DIC) constituted more than half of the leached DIC (at least from the XLHT soils; Fig. 5), whose contribution increased with re-occurring EPEs (Fig. 5). This implies that SOC mineralization during the three EPEs was underestimated by approximatey a factor of 8 when measured as $CO_2$ gas flux from soil into the column headspace only (Fig. 5). In addition, DIC loss exclusively resulting from SIC dissolution or weathering was also a significant fraction of soil carbon loss, equivalent to 219% SOC loss in the form of EPE-induced $CO_2$ during EPEs (Fig. 5). These results collectively corroborate that inorganic carbon loss is the main form of soil carbon loss in alkaline soils during EPEs.

As for the influencing factors on soil carbon leaching loss, the DIC flux was positively correlated to the amount of SIC in the soil columns and soil pH ($p < 0.05$; Fig. 6a-b). These two relationships may be self-correlated due to a positive relationship between soil pH and SIC (Liu et al., 2016). By comparison, DOC flux was linked with the amount of SOC in the soil columns, but decreased with an increasing content of silt and clay (p < 0.05; Fig. 6c). This may be explained by the stronger retention of SOC on small-sized particles with more sorption sites (Barré et al., 2014; Mayer, 1994). Interestingly, neither DOC nor DIC fluxes showed any significant relationships with the volume of leachates during EPEs (Figs. 6e-f). This indicates that we used sufficient amount of precipitation in this study to "scavenge" dissolved carbon from soils and hence these fluxes represent soil carbon's leaching potential under EPEs. Overall, total soil carbon loss through leaching under EPEs was positively related to soil pH values ($p < 0.05$; Fig. 6d), suggesting that soil pH is a critical factor determining the magnitude of soil carbon loss under EPEs.

## 3.4 Main pathways of grassland soil carbon loss under EPEs

In this study, EPE-induced soil carbon loss was composed of three parts: leachate DIC including lithogenic and biogenic DIC, leached DOC and EPE-induced $CO_2$ emission into the column headspace. Total DIC and DOC fluxes accounted for 90%, 62%, and 68% of EPE-induced total loss at XLHT, KQ, and GC, respectively, representing the major pathway of soil carbon loss in these grassland soils under EPEs. Soil carbon leaching fluxes were 25.3, 10.4, and 10.1 g C m$^{-2}$ yr$^{-1}$ in XLHT, KQ, and GC soils during three EPEs, respectively, with DIC as the dominant form in XLHT and GC soils. While DIC fluxes found for the KQ and GC soils generally fell within the range reported for grassland soils (1.3–47.8 g C m$^{-2}$ yr$^{-1}$; Parfitt et al., 1997; Brye et al., 2001; Kindler et al., 2011), the XLHT soil had a DIC flux higher than the majority ($> 50\%$) of the reported values (Fig. 7). This may be attributed to the higher SIC content and stronger dissolution of respired $CO_2$ in the XLHT soils due to its higher soil pH ($9.1 \pm 0.1$) relative to other grassland soils (pH: 5.4–7.5; Kindler et al., 2011), and the high intensity of our simulated EPEs (precipitation: 40 mm h$^{-1}$). Nonetheless, DIC fluxes in grassland soils reported in this study and elsewhere (Brye et al., 2001; Kindler et al., 2011) were significantly higher than in forest and cropland ecosystems ($p < 0.05$; Rieckh et al., 2014; Lentz and Lehrsch, 2014; Gerke et al., 2016; Herbrich et al., 2017; Siemens et al., 2012; Walmsley et al., 2011; Wang and Alva, 1999; Kindler et al., 2011), highlighting the importance of leaching as a major pathway of soil carbon loss in grasslands. By contrast, DOC fluxes in this study ($4.8 \pm 2.5$ g C m$^{-2}$) were lower than most of the reported values in forest and grassland ecosystems due to the low SOC contents in our soils (Fig. 7).

Net ecosystem production (NEP) in the temperate steppe of Inner Mongolia (XLHT and KQ) is 8.7 g C m$^{-2}$ yr$^{-1}$ (Sui and Zhou, 2013). While the EPE-induced $CO_2$ release ($2.8 \pm 0.6$ and $6.3 \pm 3.0$ g C m$^{-2}$) accounted for 32% and 72% of the NEP at XLHT and KQ, respectively; soil carbon leached during three EPEs was equivalent to 290% and 120% of NEP, with total DIC loss accounting for 244% and 33%, respectively. It is worth mentioning that biogenic DIC loss ($16.0 \pm 3.4$ g C m$^{-2}$) caused by SOC degradation accounted for 184% of NEP at XLHT, indicating the importance of biogenic DIC to leached inorganic carbon loss during EPEs. By comparison, NEP in the studied alpine grassland (68.5 g C m$^{-2}$ yr$^{-1}$; Fu et al., 2009) is much higher than in typical temperate steppe. Hence, soil carbon loss through leaching and respired $CO_2$ release accounted for 15% (DIC: 11%, DOC: 4%) and 7% of the NEP at GC, respectively. Nonetheless, the EPE-induced soil carbon loss relative to NEP was higher in this study than that estimated for grassland topsoil across Europe (12% for DIC loss, 2% for DOC loss; Kindler et al., 2011) where Net Ecosystem Exchange (NEE) reported by Kindler et al. was used as NEP according to the report of Kirschbaum et al. (2001). This was partially attributed to the lower NEP and higher SIC content in XLHT and KQ soils, underscoring that soil carbon leaching is more important in fragile ecosystems with low productivity.

An uncertainty related to the importance of leaching processes in the overall carbon budget along the "soil-river-ocean" continuum lies in the ultimate downstream fate of the leached carbon. If part of this carbon is retained in the surrounding soils or carried along from the river to the ocean in the form of DIC without outgassing into the air, it will not constitute a source of atmospheric $CO_2$ on a relatively short term (over years or decades). However, soil columns used in our study have a depth (60 cm) typical of or even deeper than the average soil depth in the alpine grasslands of Qinghai-Tibetan Plateau (Wang et al.,

2001). Hence, we assume that carbon leached in our experiments will have minimum retention in the soil. Furthermore, compared to DOC and DIC in the soil solution, the leached carbon is more likely to be subject to more intensified mineralization and outgassing during the land-ocean transfer, given more intensified mixing processes, oxygen exposure and photo-oxidation of terrestrial carbon upon releasing into the river (Hedges et al., 1997; Battin et al., 2009). Hence, we postulate

that carbon leached from soils is more vulnerable to decomposition and/or release compared to that retained in the soil. That being said, it will be necessary to confirm our results and hypothesis using field-based leaching experiments to better understand the ultimate fate of leached soil carbon: whether it will be retained in the deeper soil or show a higher degradability upon leaving the soil matrix. Such information will be complementary to our study and further elucidate the importance of leaching processes in terms of ecosystem carbon budget.

In summary, this study quantified and compared soil carbon loss through respired $CO_2$ release and leaching in three typical grassland soils of northern China under simulated EPEs. Soil $CO_2$ release was stimulated shortly after each EPE, leading to an EPE-induced $CO_2$ release equivalent to 32% and 72% of the NEP at XLHT and KQ (temperate grasslands) and 7% at GC (alpine grassland). By comparison, total soil carbon leaching fluxes accounted for 290%, 120% and 15% of the NEP at XLHT, KQ, and GC, respectively, with DIC as the main form of carbon loss in the SIC-enriched XLHT and GC soils. In view of DIC

sources, biogenic DIC loss derived from SOC mineralization contributed to more than half of the total leached DIC fluxes and accounted for 184% of the NEP at XLHT. Moreover, DIC loss increased with re-occurring EPEs in the XLHT soil with the highest pH due to increased dissolution of soil carbonates as well as elevated contribution of dissolved $CO_2$ from SOC degradation. These results also imply that SOC mineralization in alkaline grassland soils during EPEs might be underestimated, if measured only as $CO_2$ emission from soil into the atmosphere. Admittedly, our results are based on artificial soil columns

which destroyed natural soil structures, hence potentially increasing the contact between pore water and soil particles through changing soil porosity. Also, soil water content was set to be ~60% of max WHC initially in our experiment, higher than that in the field of temperate grasslands (XLHT and KQ). Thus, our measured DOC and DIC fluxes are likely to be higher than carbon leaching in the field due to greater water retention in drier soils. Hence, our estimate may represent an upper limit of soil carbon leaching potential under EPEs. Nonetheless, these results highlight that leaching loss of soil carbon, especially in

the form of DIC originated from biogenic and lithogenic carbonates, plays an important role in the regional carbon budget of grasslands located in arid and semiarid regions. Further research effort is needed to combine short-term laboratory experiments with long-term field measurements to fully assess the impacts of EPEs on soil carbon budget in these areas. In addition, with a projected increase of EPEs under climate change, soil carbon leaching processes and its influencing factors warrant better understanding and should be incorporated into soil carbon models when estimating carbon balance in grassland ecosystems.

*Data availability.* All data is available within this paper (Table 1) and in the Supplement (Dataset S1).

*Competing interests.* The authors declare that they have no conflict of interest.

## Acknowledgements

This study was supported financially by the Chinese National Key Development Program for Basic Research (2015CB954201, 2017YFC0503902), the National Natural Science Foundation of China (41422304, 31370491, 41603076), and by the International Partnership Program of Chinese Academy of Sciences (Grant No. 151111KYSB20160014). X. Feng acknowledges start-up support from the National Thousand Young Talents recruiting plan of China. We thank Dr. Jan Siemens and other anonymous reviewers for their feedback on the manuscript. The data used are listed in the tables, figures, and supplementary material of the paper.

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

**Table 1: Bulk properties of soil samples collected from the temperate and alpine grasslands for the soil column experiment (mean ± standard error; n = 3).**

| Station | Depth (cm) | SOC (%) | SIC (%) | N (%) | SOC:N ratio | pH | $\delta^{13}C$ (‰) | FWC (%) | Max WHC (%) | BD (g cm$^{-3}$) | Clay (%) | Silt (%) | Sand (%) |
|---------|-----------|---------|---------|-------|-------------|-----|---------|---------|---------|-----|------|------|------|
| Xilinhot (XLHT) | 0−20 | 1.48 ± 0.02 | 0.41 ± 0.01 | 0.18 ± 0.00 | 8.03 ± 0.18 | 8.98 ± 0.03 | −24.1 | 10.65 ± 0.11 | 47.12 ± 0.37 | 1.06 ± 0.02 | 0.4 | 64.6 | 35.0 |
|  | 20−40 | 1.00 ± 0.05 | 0.64 ± 0.00 | 0.13 ± 0.00 | 7.69 ± 0.22 | 9.09 ± 0.01 | −24.1 | 6.48 ± 0.24 | 44.92 ± 0.25 | 1.24 ± 0.05 | 0.5 | 58.2 | 41.3 |
|  | 40−60 | 0.67 ± 0.03 | 1.05 ± 0.01 | 0.09 ± 0.00 | 7.09 ± 0.22 | 9.09 ± 0.04 | −23.7 | 5.56 ± 0.11 | 39.78 ± 0.39 | 1.31 ± 0.03 | 0.6 | 58.5 | 41.0 |
| Keqi (KQ) | 0−20 | 3.36 ± 0.05 | 0.02 ± 0.00 | 0.29 ± 0.00 | 11.48 ± 0.24 | 7.79 ± 0.10 | −26.0 | 19.59 ± 0.22 | 65.57 ± 0.82 | 1.14 ± 0.03 | 0.4 | 41.0 | 58.6 |
|  | 20−40 | 2.52 ± 0.04 | 0.01 ± 0.00 | 0.22 ± 0.00 | 11.59 ± 0.27 | 7.63 ± 0.04 | −25.9 | 8.56 ± 0.05 | 53.59 ± 1.98 | 1.22 ± 0.01 | 0.2 | 55.7 | 44.1 |
|  | 40−60 | 1.65 ± 0.03 | 0.02 ± 0.00 | 0.14 ± 0.00 | 11.49 ± 0.42 | 7.57 ± 0.12 | −25.5 | 8.00 ± 0.27 | 42.92 ± 0.57 | 1.19 ± 0.01 | 0.2 | 61.6 | 38.1 |
| Gangcha (GC) | 0−20 | 3.32 ± 0.23 | 0.34 ± 0.04 | 0.31 ± 0.03 | 10.70 ± 1.28 | 8.53 ± 0.07 | −26.3 | 33.24 ± 0.68 | 60.79 ± 0.21 | n.d. | 1.3 | 75.9 | 22.8 |
|  | 20−40 | 2.90 ± 0.18 | 0.44 ± 0.10 | 0.29 ± 0.01 | 9.93 ± 0.69 | 8.60 ± 0.03 | −24.0 | 36.15 ± 0.52 | 62.03 ± 0.30 | n.d. | 0.9 | 75.8 | 23.3 |
|  | 40−60 | 2.12 ± 0.22 | 0.52 ± 0.06 | 0.20 ± 0.02 | 10.55 ± 1.50 | 8.76 ± 0.10 | −25.3 | 35.79 ± 0.91 | 62.85 ± 0.61 | n.d. | 0.6 | 64.0 | 35.4 |

SOC: soil organic carbon; SIC: soil inorganic carbon; N: nitrogen; FWC: field water content; Max WHC: maximum water holding capacity; BD: bulk density; Clay: soil particle size < 0.2 μm; Silt: 0.2 μm < soil particle size < 20 μm; Sand: soil particle size > 20 μm; n.d.: not determined.

**List of figure captions**

Figure 1: Design of the soil column experiment for monitoring soil respiration and leaching after simulated extreme precipitation events (EPEs).

Figure 2: The $\delta^{13}C$ values of respired $CO_2$ in the litter-amended Gangcha (GC) soils after the first extreme precipitation event (EPE). Mean values are shown with standard error (n = 3).

Figure 3: Total (a) and specific (b) extreme precipitation event (EPE)-induced $CO_2$ release in the litter-amended and non-amended grassland soils during three EPEs. Mean values are shown with standard deviation (n = 3). Lower-case letters (a, b, c) indicate significantly different levels among the litter-amended and non-amended soils determined by Duncan's multiple range test (one-way ANOVA, $p < 0.05$).

Figure 4: Fluxes of dissolved organic carbon (DOC) and dissolved inorganic carbon (DIC) and volume of leachates from soil columns after extreme precipitation events (EPEs). Mean values are shown with standard error (n = 3). * and ns denote significant and no difference between the litter-amended and non-amended soils determined by independent samples T test, respectively ($p < 0.05$).

Figure 5: Carbon loss fluxes from soil organic carbon (SOC) mineralization in the non-amended XLHT soils. Fluxes include extreme precipitation event (EPE)-induced $CO_2$ release and leaching of biogenic dissolved inorganic carbon (DIC), dissolved organic carbon (DOC) and lithogenic DIC. Mean values are shown with standard error (n = 3).

Figure 6: Relationship of dissolved inorganic carbon (DIC) and dissolved organic carbon (DOC) fluxes with soil properties: (a) DIC flux with total inorganic carbon in the soil columns; (b) DIC flux with soil pH; (c) DOC flux with silt and clay content of soils; (d) total soil carbon flux with soil pH; (e) DOC flux with leachate volume; (f) DIC flux with leachate volume. Mean pH values are shown with standard error (n = 3).

Figure 7: Leaching fluxes of dissolved organic carbon (DOC) and dissolved inorganic carbon (DIC) in this study compared with that reported in the literature. [1]n = 110, data from Brooks et al., 1999; Froberg et al. 2005, 2006, 2011; Gielen et al., 2011; Kindler et al., 2011; Lu et al., 2013; Michalzik et al., 2001; Sanderman et al., 2009; [2]n = 33, data from Brye et al., 2001; Kindler et al., 2011; Siemens et al., 2012; Walmsley et al., 2011; Wang and Alva, 1999; Gerke et al., 2016; Herbrich et al., 2017; Rieckh et al., 2014; Lenz, 2014; [3]n = 46, data from Brooks et al., 1999; Brye et al., 2001; Ghani et al., 2010; Kindler et al., 2011; Mctiernan et al., 2001; Parfitt et al., 2009; Sanderman et al., 2009; Tipping et al., 1999; [4]n = 8, data from Kindler et al., 2011; [5]n = 32, data from Kindler et al., 2011; Siemens et al., 2012; Walmsley et al., 2011; Wang and Alva, 1999; Gerke et al., 2016; Herbrich et al., 2017; Rieckh et al., 2014; Lenz, 2014; [6]n = 9, data from Brye et al., 2001; Kindler et al., 2011. Lower-case letters ($a_1$, $b_1$) and ($a_2$, $b_2$) represent significant different levels of DOC and

DIC fluxes in different ecosystems determined by Duncan's multiple range test, respectively, (one-way ANOVA, $p < 0.05$). Dash lines represent mean values for the investigated soils.

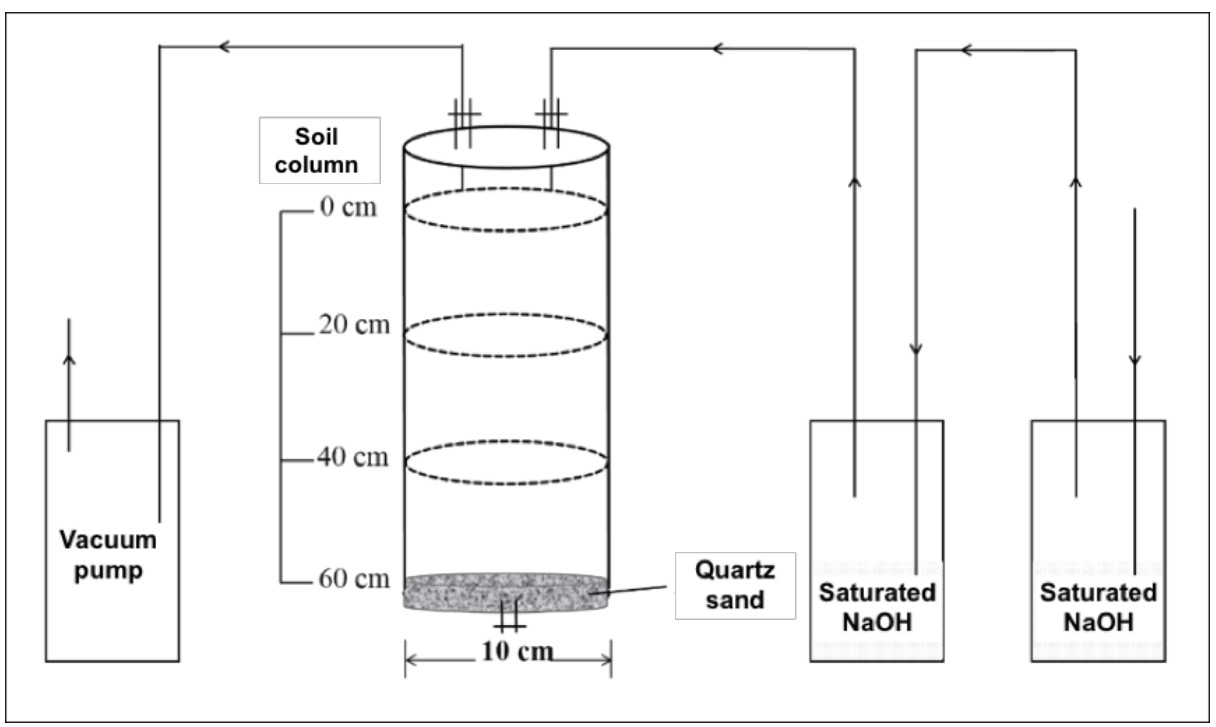

**Figure 1: Design of the soil column experiment for monitoring soil respiration and leaching after simulated extreme precipitation events (EPEs).**

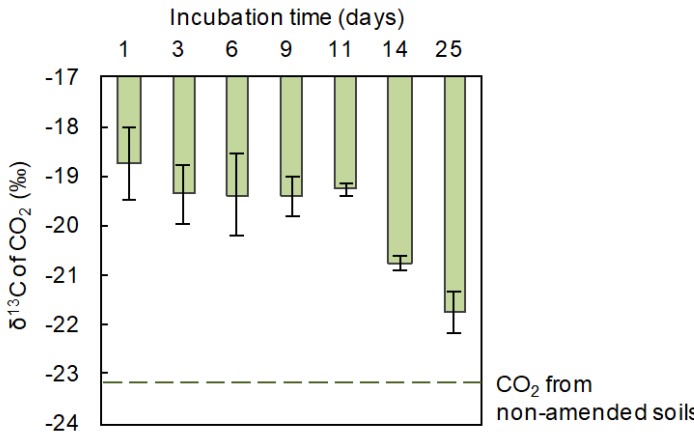

**Figure 2: The δ¹³C values of respired CO₂ in the litter-amended Gangcha (GC) soils after the first extreme precipitation event (EPE). Mean values are shown with standard error (n = 3).**

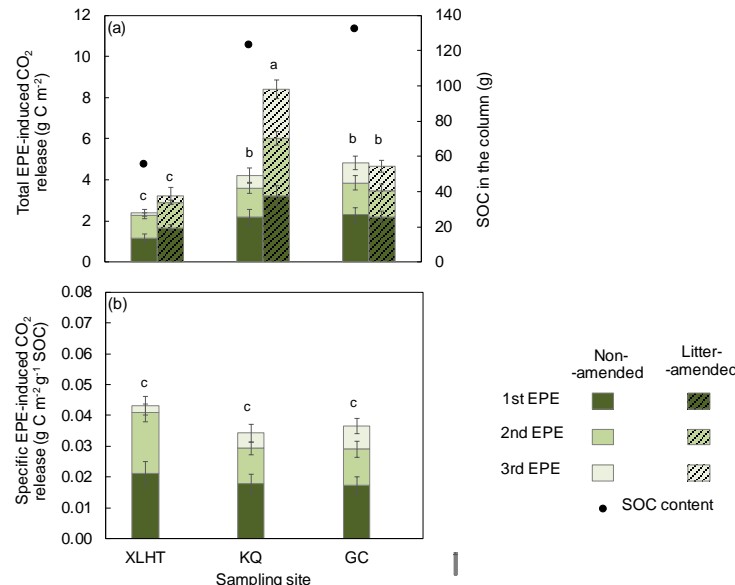

**Figure 3: Total (a) and specific (b) extreme precipitation event (EPE)-induced $CO_2$ release in the litter-amended and non-amended grassland soils during three EPEs. Mean values are shown with standard deviation (n = 3). Lower-case letters (a, b, c) indicate**
5 **significantly different levels among the litter-amended and non-amended soils determined by Duncan's multiple range test (one-way ANOVA, $p < 0.05$).**

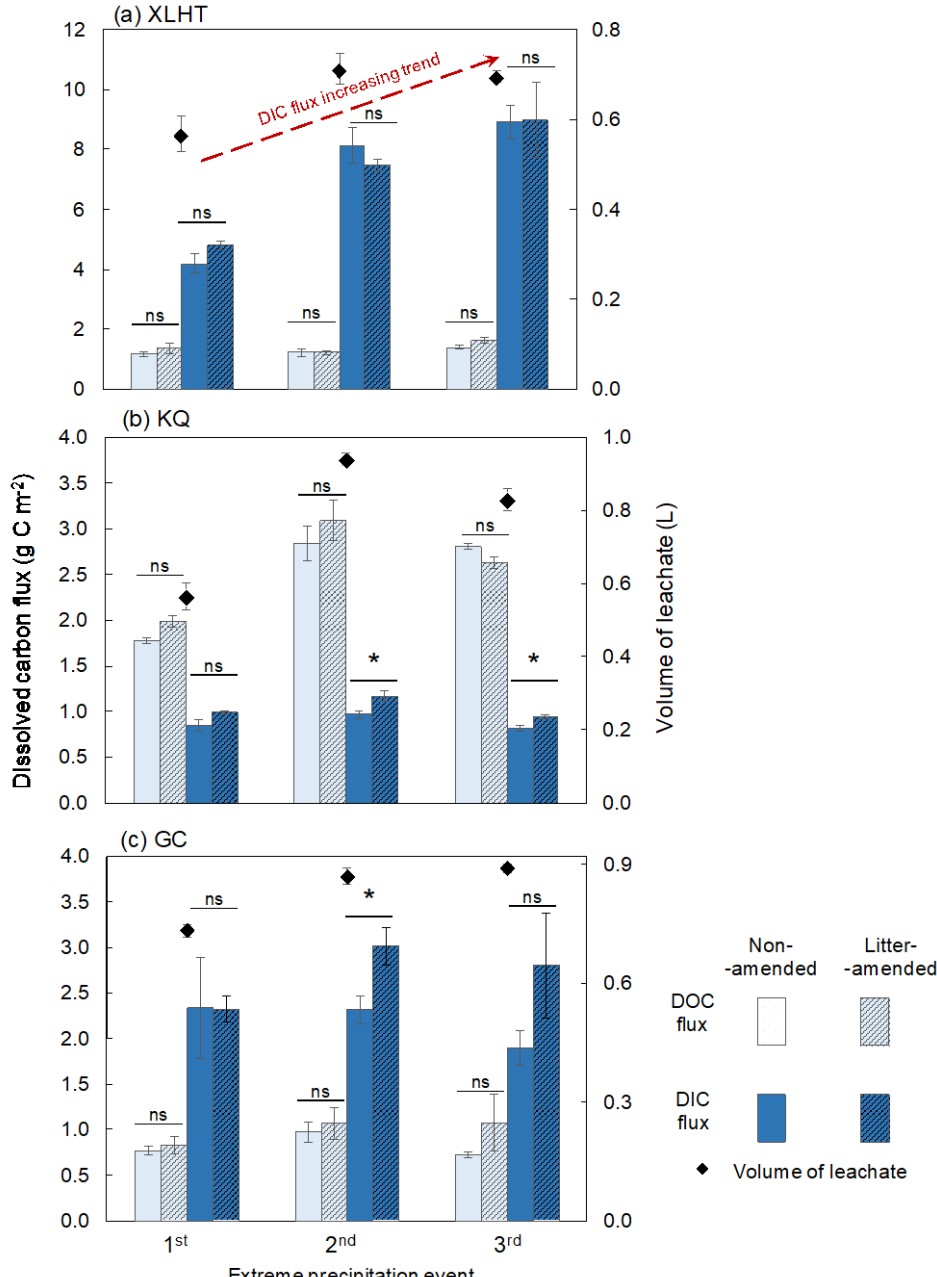

**Figure 4: Fluxes of dissolved organic carbon (DOC) and dissolved inorganic carbon (DIC) and volume of leachates from soil columns after extreme precipitation events (EPEs). Mean values are shown with standard error (n = 3). * and ns denote significant and no difference between the litter-amended and non-amended soils determined by independent samples T test, respectively ($p < 0.05$).**

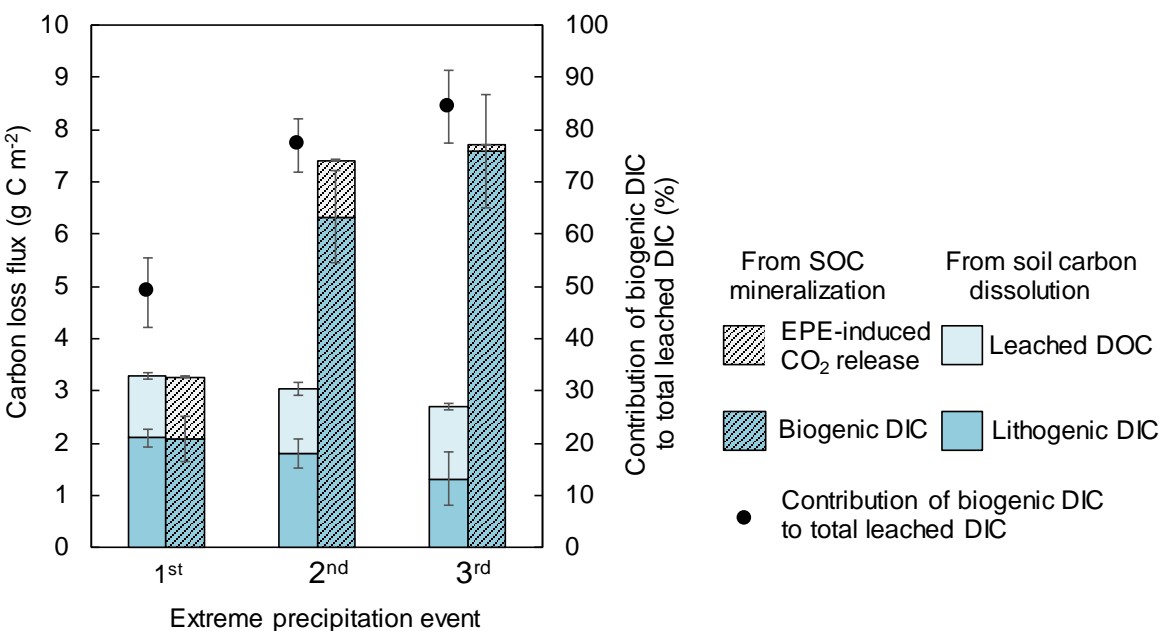

**Figure 5: Carbon loss fluxes from soil organic carbon (SOC) mineralization in the non-amended XLHT soils. Fluxes include extreme precipitation event (EPE)-induced CO$_2$ release and leaching of biogenic dissolved inorganic carbon (DIC), dissolved organic carbon (DOC) and lithogenic DIC. Mean values are shown with standard error (n = 3).**

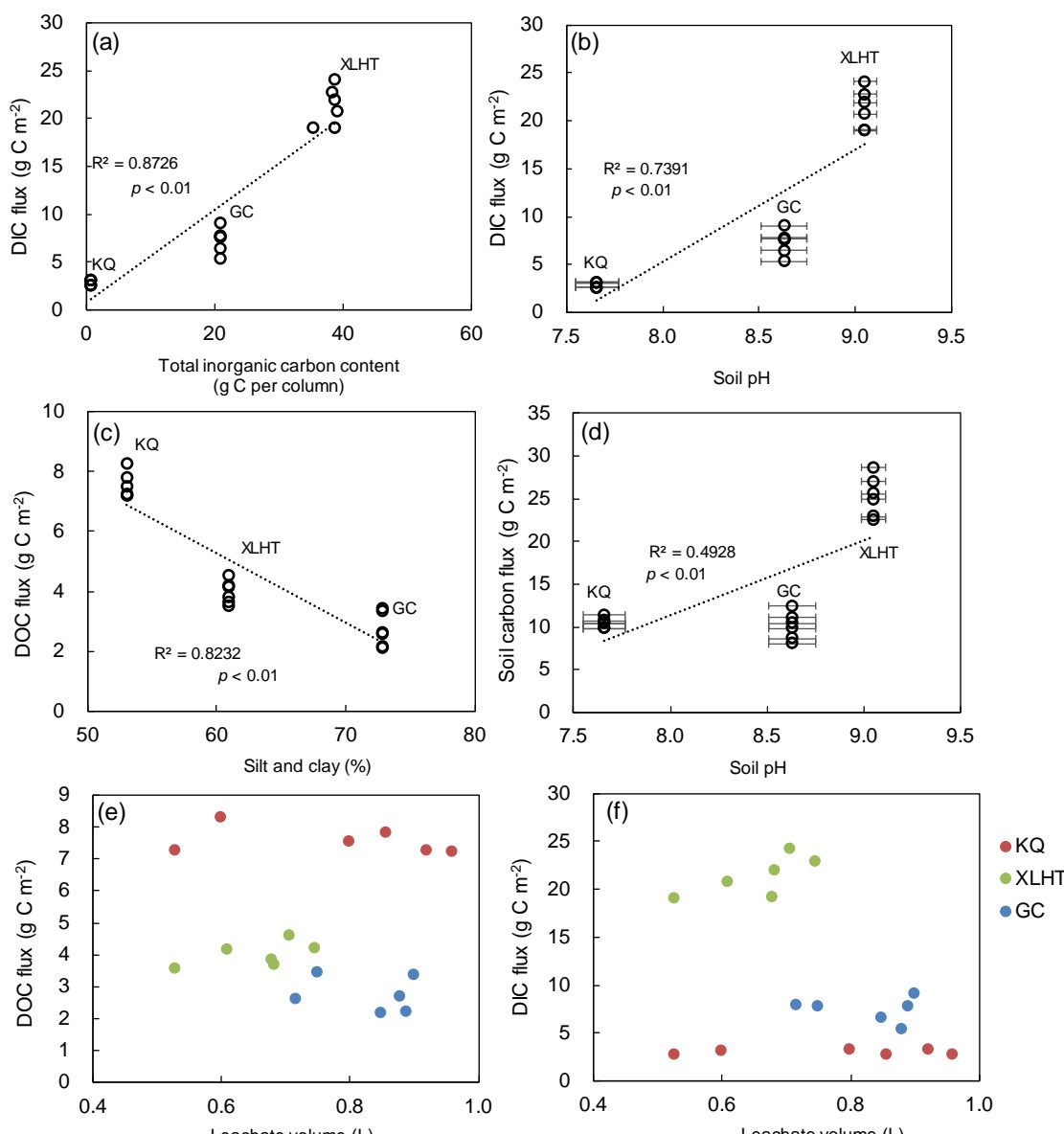

**Figure 6: Relationship of dissolved inorganic carbon (DIC) and dissolved organic carbon (DOC) fluxes with soil properties: (a) DIC flux with total inorganic carbon in the soil columns; (b) DIC flux with soil pH; (c) DOC flux with silt and clay content of soils; (d) total soil carbon flux with soil pH; (e) DOC flux with leachate volume; (f) DIC flux with leachate volume. Mean pH values are shown with standard error (n = 3).**

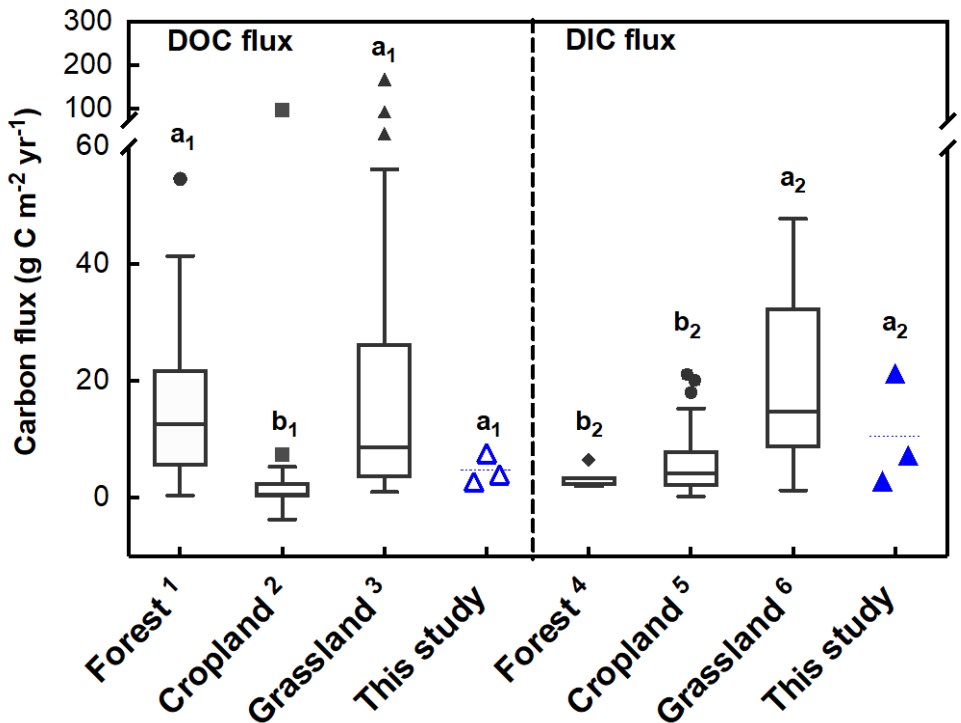

**Figure 7: Leaching fluxes of dissolved organic carbon (DOC) and dissolved inorganic carbon (DIC) in this study compared with that reported in the literature. [1]n = 110, data from Brooks et al., 1999; Froberg et al. 2005, 2006, 2011; Gielen et al., 2011; Kindler et al., 2011; Lu et al., 2013; Michalzik et al., 2001; Sanderman et al., 2009; [2]n = 33, data from Brye et al., 2001; Kindler et al., 2011; Siemens et al., 2012; Walmsley et al., 2011; Wang and Alva, 1999; Gerke et al., 2016; Herbrich et al., 2017; Rieckh et al., 2014; Lenz, 2014; [3]n = 46, data from Brooks et al., 1999; Brye et al., 2001; Ghani et al., 2010; Kindler et al., 2011; Mctiernan et al., 2001; Parfitt et al., 2009; Sanderman et al., 2009; Tipping et al., 1999; [4]n = 8, data from Kindler et al., 2011; [5]n = 32, data from Kindler et al., 2011; Siemens et al., 2012; Walmsley et al., 2011; Wang and Alva, 1999; Gerke et al., 2016; Herbrich et al., 2017; Rieckh et al., 2014; Lenz, 2014; [6]n = 9, data from Brye et al., 2001; Kindler et al., 2011. Lower-case letters ($a_1$, $b_1$) and ($a_2$, $b_2$) represent significant different levels of DOC and DIC fluxes in different ecosystems determined by Duncan's multiple range test, respectively, (one-way ANOVA, $p < 0.05$). Dash lines represent mean values for the investigated soils.**