# Peer review of "Comparing soil carbon loss through respiration and leaching under extreme precipitation events in arid and semi-arid grasslands"

_Biogeosciences, 2017_

## Referee Comment (RC1) · Anonymous Referee #1 · 24 Aug 2017

General Comments

The manuscript of Liu et al presents very interesting information regarding the triggering of soil carbon losses via respiration and leaching by extreme precipitation events. The results of the soil column experiments illustrate that leaching losses of carbon from soils as consequence of extreme precipitation events may well exceed carbon losses due to enhanced respiration. However, the overall relevance of dissolved organic and dissolved inorganic carbon leaching losses will ultimately depend on the fate of the leached carbon on its way to groundwater and further through rivers into the ocean. If the dissolved organic carbon and inorganic carbon are retained in subsoils, then the

leached C might well be finally emitted from the soil to the atmosphere in the form of $CO_2$, if the dissolved organic carbon (DOC) is mineralized or if the soil water is lost via evapotranspiration, thus releasing the dissolved inorganic carbon (DIC)...This aspect of the importance of the downstream fate of leached carbon for the overall relevance of the leaching pathway for the carbon balance is missing in the manuscript.

When judging the relevance of dissolved inorganic carbon leaching for the carbon balance, it is also crucial to differentiate between the biogenic fraction of DIC and its lithogenic (carbonate-rock derived) fraction. In my opinion, it is much more straight forward to compare the biogenic leaching losses of DIC with the NEP, than total DIC leaching losses. The authors might want to consider this in their discussion of their results in lines 25ff on page 9. In this context the application of the isotopic mass balance model is important. The results of this model depend strongly on the delta 13C values of the end-members carbonate and $CO_2$ from soil respiration. The authors decided to use the delta 13C of the soil organic matter of -24 per-mille to calculate the biogenic fraction of dissolved inorganic carbon. Because isotopic fractionation occurs during the mineralization of soil organic matter, the authors might additionally use their delta 13C value of -23.1 per-mille as end member in order to assess the uncertainty that is associated with potential isotopic fractionation during mineralization and diffusive $CO_2$ transport in soil (Cerling et al., 1991. On the isotopic composition of carbon in soil carbon dioxide. Geochim. Cosmochim. Acta 55, 3403-3405). Quantitatively more important than the isotopic fractionation during mineralization and diffusion of $CO_2$ for the delta 13C value used as end member for the soil organic carbon derived fraction of DIC is the isotopic fractionation between $CO_2$ in the gas phase and bicarbonate (Zhang et al., 1995. Carbon isotope fractionation during gas-water exchange and dissolution of $CO_2$. Geochim. Cosmochim. Acta 59, 107-114). In the pH range of the investigated soils, the vast majority of the DIC will be present as bicarbonate ($HCO_3^-$). According to Zhang et al. (1995), isotope fractionation between the gas phase and the aqueous phase will shift the delta 13C of bicarbonate in equilibrium with gaseous $CO_2$ by some 10-11 per-mille. Hence, the end member delta 13C of DIC in equilibrium with $CO_2$,

which has a delta 13C value of -24 per-mille, can be around -14 to -13 per-mille. Considering this isotopic fractionation between gaseous CO2 and bicarbonate will greatly increase the calculated fractions of biogenic (soil organic carbon-derived) DIC.

Specific comments On page 3, line 19, the soil is classified as "chestnut soil". This classification is not in line with the international soil classification system of the World Reference Base for Soil Resources (WRB, 2015). Please classify your soils also according to the WRB system (http://www.fao.org/3/a-i3794e.pdf). On page 6, lines 20ff, the authors argue that the variation in SIC contents between sites are caused by the variation of pH values, suggesting a causality between pH (independent variable) and SIC content (dependent variable). The question is, whether the pH is really controlling SIC contents or vice versa...

On page 9, lines 29ff, the authors cite Kindler et al. (2011) for numbers of DIC leaching losses equaling 12% of NEP and DOC leaching losses equaling 2% of NEP. I do not understand how the authors extracted these numbers from the Kindler et al. (2011) publication.

Starting on page 9, line 31, the authors argue that the carbon loss due to extreme precipitation events was much greater than carbon losses through warming-enhances respiration. This comparison is perhaps misleading, because it implies that extreme precipitation events occur only as consequence of climate change. More correct would be the comparison of carbon losses due to warming-enhanced respiration with carbon losses due to "climate change-enhanced" extreme precipitation events.

---

## Author Comment (AC1) · 18 Sep 2017

**Response to Anonymous Referee #1**

We appreciate Referee #1's critical and detailed assessment of our manuscript and we are grateful for his/her constructive comments which helped us to greatly refine our paper. Here we provide a point-to-point response to all the issues raised by the referee. We hope our replies and revisions will satisfy all the requests.

**General Comments:**

**Comment 1:**

The manuscript of Liu et al presents very interesting information regarding the triggering of soil carbon losses via respiration and leaching by extreme precipitation events. The results of the soil column experiments illustrate that leaching losses of carbon from soils as consequence of extreme precipitation events may well exceed carbon losses due to enhanced respiration. However, the overall relevance of dissolved organic and dissolved inorganic carbon leaching losses will ultimately depend on the fate of the leached carbon on its way to groundwater and further through rivers into the ocean. If the dissolved organic carbon and inorganic carbon are retained in subsoils, then the leached C might well be finally emitted from the soil to the atmosphere in the form of $CO_2$, if the dissolved organic carbon (DOC) is mineralized or if the soil water is lost via evapotranspiration, thus releasing the dissolved inorganic carbon (DIC)...This aspect of the importance of the downstream fate of leached carbon for the overall relevance of the leaching pathway for the carbon balance is missing in the manuscript.

**Response:**

This is a very good point! The downstream fate of the leached carbon is indeed very important as it will ultimately determine the relevance of leaching processes to the overall carbon budget or balance along the "soil-river-ocean" continuum. If part of the leached carbon is retained in the deeper soils or transformed and carried along from the river to the ocean in the form of DIC, it will not constitute a source of atmospheric $CO_2$ on a relatively short term (over years or decades). However, compared to DOC and DIC in the soil solution, the leached carbon is more likely to be subject to more intensified mineralization and outgassing during the land-ocean transfer, given more intensified mixing processes, oxygen exposure and photo-oxidation of terrestrial carbon upon releasing into the river (Hedges et al., 1997; Battin et al., 2009). Hence, we postulate that carbon leached from soils is more vulnerable to decomposition and/or release compared to that retained in the soil. That being said, it will be necessary to confirm our results and hypothesis using field-based leaching experiments to better understand the ultimate fate of leached soil carbon: whether it will be retained in the deeper soil or show a higher degradability upon leaving the soil matrix. Such information will be complementary to our study and further elucidate the importance of leaching processes in terms of ecosystem carbon budget.

The above considerations and discussions are now added as a separate paragraph in Section 3.4 in the revised paper:

> "An uncertainty related to the importance of leaching processes in the overall carbon budget along the "soil-river-ocean" continuum lies in the ultimate downstream fate of the leached carbon. If part of this carbon is retained in the surrounding soils or carried along from the river to the ocean in the form of DIC without outgassing into the air, it will not constitute a source of atmospheric $CO_2$ on a relatively short term (over years or decades). However, soil columns used in our study has a depth (60 cm) typical of or even deeper than the average soil depth in the alpine grasslands of Qinghai-Tibetan Plateau

*(Wang et al., 2001). Hence, we assume that carbon leached in our experiments will have minimum retention in the soil. Furthermore, compared to DOC and DIC in the soil solution, the leached carbon is more likely to be subject to more intensified mineralization and outgassing during the land-ocean transfer, given more intensified mixing processes, oxygen exposure and photo-oxidation of terrestrial carbon upon releasing into the river (Hedges et al., 1997; Battin et al., 2009). Hence, we postulate that carbon leached from soils is more vulnerable to decomposition and/or release compared to that retained in the soil. That being said, it will be necessary to confirm our results and hypothesis using field-based leaching experiments to better understand the ultimate fate of leached soil carbon: whether it will be retained in the deeper soil or show a higher degradability upon leaving the soil matrix. Such information will be complementary to our study and further elucidate the importance of leaching processes in terms of ecosystem carbon budget."*

References:

Battin, T. J., Luyssaert, S., Kaplan, L. A., Aufdenkampe, A. K., Richter, A., and Tranvik, L. J.: The boundless carbon cycle, Nature Geoscience, 2, 598-600, 2009.

Hedges, J. I., Keil, R. G., and Benner, R.: What happens to terrestrial organic matter in the ocean?, Org. Geochem., 27, 195-212, 1997.

Wang, S.Q., Zhu, S.L., and Zhou, C.H.: Characteristics of spatial variability of soil thickness in China, Geographical research, 20, 161-169, 2001.

**Comment 2:**

When judging the relevance of dissolved inorganic carbon leaching for the carbon balance, it is also crucial to differentiate between the biogenic fraction of DIC and its lithogenic (carbonate-rock derived) fraction. In my opinion, it is much more straight forward to compare the biogenic leaching losses of DIC with the NEP, than total DIC leaching losses. The authors might want to consider this in their discussion of their results in lines 25ff on page 9. In this context the application of the isotopic mass balance model is important. The results of this model depend strongly on the delta 13C values of the end-members carbonate and CO2 from soil respiration. The authors decided to use the delta 13C of the soil organic matter of -24 per-mille to calculate the biogenic fraction of dissolved inorganic carbon. Because isotopic fractionation occurs during the mineralization of soil organic matter, the authors might additionally use their delta 13C value of -23.1 per-mille as end member in order to assess the uncertainty that is associated with potential isotopic fractionation during mineralization and diffusive CO2 transport in soil (Cerling et al., 1991. On the isotopic composition of carbon in soil carbon dioxide. Geochim. Cosmochim. Acta 55, 3403-3405). Quantitatively more important than the isotopic fractionation during mineralization and diffusion of CO2 for the delta 13C value used as end member for the soil organic carbon derived fraction of DIC is the isotopic fractionation between CO2 in the gas phase and bicarbonate (Zhang et al., 1995. Carbon isotope fractionation during gas-water exchange and dissolution of CO2. Geochim. Cosmochim. Acta 59, 107-114). In the pH range of the investigated soils, the vast majority of the DIC will be present as bicarbonate (HCO3-). According to Zhang et al. (1995), isotope fractionation between the gas phase and the aqueous phase will shift the delta 13C of bicarbonate in equilibrium with gaseous CO2 by some 10-11 per-mille. Hence, the end member delta 13C of DIC in equilibrium with CO2, which has a delta 13C value of -24 per-mille, can be around -14 to -13 per-mille. Considering this isotopic fractionation between gaseous CO2 and bicarbonate will greatly increase the calculated fractions of biogenic (soil organic carbon-derived) DIC.

**Response:**

We must thank the reviewer for pointing out an excellent point that we overlooked. The dissolution of $CO_2$ produced by SOC degradation does cause large isotope fractionation on the biogenic carbonate/bicarbonate. Taking this into account, we have revised our endmember values ($\delta^{13}C_{biogenic-DIC}$ is estimated to be $-16‰$), and revised the following parts:

Section 2.4:

*"The relative contribution of lithogenic carbonate and biogenic DIC derived from SOC degradation to leached DIC was assessed according to the following isotopic mass balance model:*

$$f_{carbonate} + f_{biogenic-DIC} = 1 \tag{1}$$

$$f_{carbonate} \times \delta^{13}C_{carbonate} + f_{biogenic-DIC} \times \delta^{13}C_{biogenic-DIC} = \delta^{13}C_{DIC} \tag{2}$$

*where $f_{carbonate}$ and $f_{biogenic-DIC}$ are proportion of carbonate- and biogenic DIC in total DIC; $\delta^{13}C_{carbonate}$ is the $\delta^{13}C$ value of soil carbonate, equivalent to 0‰ (Edwards and Saltzman, 2016); and $\delta^{13}C_{biogenic-DIC}$ is the $\delta^{13}C$ value of biogenic carbonate/bicarbonate derived from the dissolution of $CO_2$ produced by SOC degradation, which is estimated to shift by approximately 8‰ compared with the $\delta^{13}C$ value of soil-respired $CO_2$ ($-24‰$ here) due to isotope fractionation during $CO_2$ dissolution (Zhang et al., 1995). Hence, $\delta^{13}C_{biogenic-DIC}$ is estimated to be $-16‰$. $\delta^{13}C_{DIC}$ is the measured $\delta^{13}C$ signature of leached DIC. According to Hendy (1971) and Doctor et al. (2008), isotopic fractionation of leached DIC due to $CO_2$ loss in an open system is insignificant when the partial pressure of $CO_2$ ($pCO_2$) in the solution is lower than twice that of the surrounding atmosphere. Therefore, due to the much lower $pCO_2$ in the XLHT leachates (~ 200 μatm; Table S2) compared to that in the ambient atmosphere (> 400 μatm), the influence of $CO_2$ outgassing on the $\delta^{13}C$ of leached DIC was not considered in the present study."*

Section 3.3:

*"Based on the isotopic mass balance Eq. (1) and (2), lithogenic carbonate (with a $\delta^{13}C$ value of 0‰) contributed 51.2% to the leached DIC while biogenic DIC produced by SOC degradation contributed 48.4% (Fig. 5). The $\delta^{13}C$ value of leached DIC decreased to $-12.3‰$ and $-13.5‰$ during the second and third EPEs, corresponding to a contribution of 77.0% and 84.4% by biogenic sources in the total DIC, respectively (Fig. 5). These results confirm our previous hypothesis that SOC decomposition contributed significantly to soil DIC fluxes. Combined with the total flux rate, we calculated that both lithogenic and biogenic DIC fluxes were ~2.1 g C m$^{-2}$ in the first EPE. Subsequently, lithogenic DIC flux decreased to ~1.3 g C m$^{-2}$ while biogenic DIC flux increased to 7.6 g C m$^{-2}$ in the third EPE. This demonstrates that increased SOC degradation mainly contributed to the increased DIC fluxes with repeated EPEs."*

Section 3.4:

*"It is worth mentioning that biogenic DIC loss (16.0 ± 3.4 g C m$^{-2}$) caused by SOC degradation accounted for 184% of NEP at XLHT, indicating the importance of biogenic DIC to SIC loss during EPEs."*

Reference:

Zhang, J., Quay, P. D., and Wilbur, D. O.: Carbon isotope fractionation during gas-water exchange and dissolution of $CO_2$, Geochim. Cosmochim. Acta, 59, 107-114, http://dx.doi.org/10.1016/0016-7037(95)91550-D, 1995.

**Specific Comments:**

**Comment 3:**

On page 3, line 19, the soil is classified as "chestnut soil". This classification is not in line with the international soil classification system of the World Reference Base for Soil Resources (WRB, 2015). Please classify your soils also according to the WRB system (http://www.fao.org/3/a-i3794e.pdf).

**Response:**

Revised. Soils in Xilinhot and Keqi are classified as Calcic Chernozems according to the World Reference Base for Soil Resources (Steffens et al., 2008; IUSS working group WRB, 2015) while soils in Gangcha are mainly Gelic Cambisol (IUSS working group WRB, 2015).

**Comment 4:**

On page 6, lines 20ff, the authors argue that the variation in SIC contents between sites are caused by the variation of pH values, suggesting a causality between pH (independent variable) and SIC content (dependent variable). The question is, whether the pH is really controlling SIC contents or vice versa. . .

**Response:**

SIC content is related to parent materials as well as soil pH. However, in the present study regions, soil pH is the key factor controlling SIC variation across the Mongolian and Tibetan grasslands according to the results of Shi et al. (2012). We have added one sentence to clarify this:

> *"This dependence of SIC content on soil pH is consistent with the results of Shi et al. (2012), showing that pH is the most important factor controlling SIC variation across the Mongolian and Tibetan grasslands."*

**Comment 5:**

On page 9, lines 29ff, the authors cite Kindler et al. (2011) for numbers of DIC leaching losses equaling 12% of NEP and DOC leaching losses equaling 2% of NEP. I do not understand how the authors extracted these numbers from the Kindler et al. (2011) publication.

**Response:**

This is clarified now in the revised paper:

According to the report of Kirschbaum et al. (2001), both Net Ecosystem Exchange (NEE) and NEP refer to net primary production minus carbon loss from heterotrophic respiration, $R_h$:

$$NEE = NEP = NPP - R_h = GPP - R_{autotrophic\ respiration} - R_h$$

These terms are used somewhat interchangeably, with NEE used more often when they are addressed based on measurements of gas exchange rates using atmospheric measurements over the time scales of hours, whereas NEP is more often used if measurements are based on ecosystem-carbon stock changes, typically measured over a minimal period of one year.

The NEE in Kindler et al. (2011) equals to gross primary productivity minus ecosystem respiration, excluding C deprivation with harvest, fires, etc. Therefore, we extracted NEE in Table 5 of Kindler et al. (2011) as NEP for calculating the proportion of leaching DOC and DIC in NEP. Relevant description has been added in Section 3.4 as follows:

> *"Nonetheless, the EPE-induced soil carbon loss relative to NEP was higher in this study than that*

*estimated for grassland topsoil across Europe (12% for DIC loss, 2% for DOC loss; Kindler et al.,*
*2011) where Net Ecosystem Exchange (NEE) reported by Kindler et al. was used as NEP according to*
*the report of Kirschbaum et al. (2001)."*

**Comment 6:**

Starting on page 9, line 31, the authors argue that the carbon loss due to extreme precipitation events was much greater than carbon losses through warming-enhances respiration. This comparison is perhaps misleading, because it implies that extreme precipitation events occur only as consequence of climate change. More correct would be the comparison of carbon losses due to warming-enhanced respiration with carbon losses due to "climate change-enhanced" extreme precipitation events.

**Response:**

The reviewer raised a good point. It is not fair to compare carbon leaching through all annual EPEs with warming-induced respiration increase. To be more consistent and robust, we decide to delete the discussion on the comparison of carbon losses due to warming-induced respiration with carbon losses due to annual EPEs.

---

## Referee Comment (RC2) · Anonymous Referee #2 · 12 Oct 2017

General comments The manuscript presents a study that attempt to evaluate the effect of extreme precipitation events on soil carbon losses in arid and semi-arid grasslands. The objective was to distinguish between C losses due to respiration and leaching. Additionally, leaching losses were separated into DIC and DOC losses. Therefore, a soil column experiment was conducted, were respiration and leaching losses were measured after an artificial precipitation events. Soil inorganic carbon losses due to leaching was higher than due to an enhanced respiration. As already mentioned by the first referee, the relevance of C losses depends on the fate of DIC and this should be more pronounced in the discussion. In addition, soil carbon losses due to DIC leaching has to be more discussed in detail, especially the fact that about 50% (or even

more with the already recalculated values) of the DIC originates from SOC degradation (dissolved CO2). In consequence, the conclusion that most soil carbon during EPE is lost due to DIC (in partical SIC), might be not true. On the contrary, most of the DIC originates from dissolution of CO2, which originates from SOC mineralization and not SIC leaching. This should be discussed much more in detail.

The experimental setup seems appropriate for the objectives presented in the manuscript, however the presented results need some reconsideration and re-calculation, especially respiration data should presented as specific respiration to account for different SOC contents in the investigated soils (for more detail see specific comments).

Specific comments

Page 3,line 25: The dimension of the soil pits seems quiet small 10 cm x 10 cm. Even by using a shovel I would expect that you need a bigger area to go down to 70 cm.

Page 5, line 1ff: How was the water added to the soil columns? Did you had needles in the top lid of the soil column? Did you used a constant rate, like 0.5 mm per minute? How much time was in between the EPE events? How long did you wait until you start a new EPE? How where the soil columns treated in between the CO2 measurements? Were they closed or flushed with constant air flow? Please provide more information about the experimental setup for the reader.

Page 7, line 8ff: Additionally, to the respiration rate I'd suggest to calculate a specific respiration rate, which is the respiration rate divided by the amount of SOC (mg CO2-C g-1 SOC h-1). This would allow an easier comparison of the different soils with different SOC content.

Page 7 line 13f: The authors argue that litter addition increase respiration due to miner-alization of labile litter compounds and priming effects. However, here were no values presented which would underline this statement. Since the authors measured $\delta$13CO2,

they should be able to separate litter mineralization from SOC mineralization.

Page 7, line 17ff: The used mixing model should be mentioned in the method section. Further with the given isotopic values I can not understand how the authors calculated contribution of litter mineralization to total respiration. Using a mixing model of: 1 – ((c_mix-c_litter)/(c_control-c_litter)), where c_mix is the isotopic value of CO2 from the litter amended sample (-18,7‰, c_litter the isotopic value of the added litter (-16,2‰ and c_control the isotopic value of CO2 from the non-amended sample (-23,1‰, the contribution of litter mineralization to total respiration was around 64% at day 1 and only 19% at day 25. Which values for $\delta$13CO2 values did you used for the control (non-amended) samples. Did you measured $\delta$13CO2 for the control only at the beginning or at the same resolution as $\delta$13CO2 for the litter-amended samples? Further, are there any isotopic measurements of the other two sites. If so, why they are not shown here?

Page 7 line21ff.: Despite the fact that the calculation described here might be simple, it should be part of the method section and not of the result/discussion section. "EPE-induced CO2 release was higher in the KQ and GC soils than in the XLHT soil (p < 0.05; Fig. 3) that had a lower SOC content and a lower SOC:N ratio (Table 1)", as mentioned above, I suggest to calculate a specific respiration normalized to the absolute amount of SOC in the soil column. The specific respiration will provide more information about the stability and the loss of C from the different sites. A rough calculation based on figure 3 revealed that respiration of the 3 sites in the non-amended treatment might not differ. However, this has to be checked with the measurement values. It is also not clear to which EPE is shown in Fig 3, is it the first, second or third one?

Page 7, line 25ff: "Litter amendment significantly increased the EPE-induced CO2 release from the KQ soil (p < 0.05) but did not have any effect on the XLHT and GC soils." - I disagree with this statement, because figure 3 only show total CO2 release which is the sum of CO2 from litter mineralization and SOC mineralization from the mineral soil. Therefore, it is only possible to say that EPE increased total CO2 release. However, as mentioned above, I would recommend to use the isotopic data of CO2 to separate

respiration into the two sources litter and soil. Which would allow a comparison of SOC mineralization between non-amended and amended soils.

Further the authors explained the higher CO2 release in the KQ soil with a coarser soil texture and less sorptive protection of DOC, however I can't follow this explanation since Table 1 show a lower sand content in the KQ than in the XLHT soils. "These results suggest that soil texture, SOC content and quality are important factors influencing EPE-enhanced soil respiration."This is a quite strong statement, because there is no information given about the SOC quality in the different soils used in this study, also the soil texture of the used soils is more or less similar and the presented results does not justify such a strong statement. In fact, the higher CO2 release in the KQ can be explained by several other factors, such as the higher pH of the GC and XLHT soils. The higher the pH the more CO2 is dissolved, which can be seen in Fig 4 in the higher DIC contents of these sites. Also negative priming effects may occur in the GC and XLHT sites which lead to a reduced SOC mineralization, because microorganisms utilize the added litter first. This should be discussed in more detail.

Page 8 line 8ff: The authors mentioned that CO2 from litter decomposition contributed to DIC, however in the following lines the argue that this effect is more pronounced in the KQ soils than in the XLHT and GC, due to the higher SIC content of the later once. This is true speaking in relative amounts, but considering absolute amounts I would expect that much more CO2 is dissolved in the GC and XLHT soils due to the higher pH Therefore CO2 might also be a source of DIC in these soils with high SIC content. Which is also shown in line 23-24 on the same page.

Page 8, line 19f.: Why was this measurement only done for the XLHT soils? Or why the results from the other sites not shown?

Page 8, line 27f.: "Interestingly, increasing DIC fluxes were not observed in the KQ and GC soils (Fig. 4), although they had higher SOC content and degradation (i.e., respiration) rates (Fig. S5)" - What do you mean with higher degradation?

Page 8, line 31ff.: Here the authors calculated a specific leaching loss normalized to SIC content and compared it to DOC losses. However, in line 9 same page, it is mentioned that especially for the soils with low SIC content, the dissolution CO2 from SOC and litter mineralization is a major source for DIC. Therefore, the high value of 32.5 mg C g-1 SIC has to attributed to SOC mineralization and not to SIC loss. In consequence, the drawn conclusion that SIC loss is the main form of soil carbon loss in neutral to alkaine soils might be questioned, at least for the KQ soils.

Page 9, line 33: "It is also worth mentioning that soil carbon leaching fluxes in this study (10.1−25.3 g C m −2 yr −1 ) far exceed annual SOC loss through warming-enhanced respiration at these sites (0.2−0.6 g C m −2 yr −1 ) given an assumed temperature sensitivity of 2 in climate..." - I think the authors can not make such a comparison, since most of the DIC in this study originates from SOC degradation, therefore DIC losses has to be attributed to respiration losses and not DIC losses. Meaning that dissolution of CO2 needs first mineralization of SOC by microorganism.

Technical corrections Fig. S4 and S5 are swapped, the caption of Fig. S4 belongs to Fig. S5 and vice versa.

---

## Author Response (AR1)

Dear Editor,

Thank you for allowing us to submit a revised version of our manuscript. We greatly appreciate the detailed and constructive comments of you and two reviewers which helped us to improve the manuscript. We addressed all comments as attached below (highlighted in red for reviewer 1's comments and in blue for reviewer 2's comments) and hope that we adequately solved the requests.

With kind regards,

Xiaojuan Feng

(We attached a marked version of the manuscript and the supplement with changes highlighted at the end of this pdf.)

**Response to Editor**

**Comment:**

(1) Your manuscript has now been seen by two reviewers, who both provided a very thorough assessment and gave several excellent suggestions to improve your work. From your reply, it appears that you will be able to address all their concerns. However, the required changes will alter the manuscript substantially. As such, I invite you to re-submit your manuscript, after which it will be sent out for a second review round.

(2) p.s. please note that your answer to comment 6 by reviewer 2 still needs a little work. At the start of the paragraph, you state that "Total respired CO2 was higher in the litter-amended than non-amended soils before and after EPEs (Fig. S3), likely caused by the degradation of labile components in the fresh litter and/or increased degradation of SOC primed by litter additions (Fröberg et al., 2005; Ahmad et al., 2013)." However, towards the end of the paragraph you show that litter addition actually decreased the decomposition of native SOC. The first sentence clearly suggests the opposite, which makes the text a bit confusing.

**Response:**

(1) We appreciate reviewers' critical and detailed assessment of our manuscript and we are grateful for his/her constructive comments which helped us to greatly refine our paper. Here we provide a point-to-point response to all the issues raised by the referee. We hope our replies and revisions will satisfy all the requests.

(2) Thank you very much for your kind suggestions. In this paragraph, we firstly would like to provide two possible reasons responsible for the higher total respired $CO_2$ in litter-amended than non-amended soils. However, whether and how these two reasons worked was not sure until our below isotope analysis. To avoid any possible confusion, we have revised this paragraph as follows:

*"Total respired $CO_2$ was higher in the litter-amended than non-amended soils before and after EPEs (Fig. S3), likely caused by one or two following reasons: (1) the degradation of labile components in the fresh litter; (2) increased degradation of SOC primed by litter additions (Fröberg et al., 2005; Ahmad et al., 2013). These two reasons might affect $CO_2$ release in litter-amended soils in two ways: (1) both of them had positive effects; (2) one of them had positive effects and the other one had negative ones, but the total net effects was positive. To distinguish the influences of above two reasons on total respired $CO_2$ and further differentiate the contribution of litter (C4) and SOC (C3) to the respired $CO_2$, we examined the $\delta^{13}C$ values of $CO_2$ evolved from the GC soils after the first EPE. On the first day after EPE, $CO_2$ from the non-amended and litter-amended GC soils had a $\delta^{13}C$ value of –23.1‰ and –18.7‰, respectively. The latter was close to the $\delta^{13}C$ signature of the added litter (–16.25‰). Using the two-endmember mixing model of Eq. (1) and (2), we calculated that litter contributed ~64% of the respired $CO_2$ in the litter-amended GC soils. However, along with the consumption of labile carbon in litter, the $\delta^{13}C$ signature of $CO_2$ decreased from –18.7‰ on Day 1 to –21.8‰ on Day 25 after EPE in the litter-amended soils (Fig. 2). Accordingly, the proportion of litter-derived $CO_2$ decreased from 64% to 20%. The litter-derived $CO_2$ flux in litter-amended GC soils was estimated to range from 7.0 to 17.5 mg C $m^{-2} h^{-1}$, while the SOC-derived $CO_2$ flux increased from 6.2 to 15.7 mg C $m^{-2} h^{-1}$ after the first EPE (Fig. S3). Compared with the SOC-derived $CO_2$ flux in non-amended GC soils (ranging from 17.2 to 27.1 mg C $m^{-2} h^{-1}$), litter addition had a negative priming effect on the degradation of native SOC while increasing total respiration through labile litter degradation."*

**Response to Referee #1**

**General Comments:**

**Comment 1:**

The manuscript of Liu et al presents very interesting information regarding the triggering of soil carbon losses via respiration and leaching by extreme precipitation events. The results of the soil column experiments illustrate that leaching losses of carbon from soils as consequence of extreme precipitation events may well exceed carbon losses due to enhanced respiration. However, the overall relevance of dissolved organic and dissolved inorganic carbon leaching losses will ultimately depend on the fate of the leached carbon on its way to groundwater and further through rivers into the ocean. If the dissolved organic carbon and inorganic carbon are retained in subsoils, then the leached C might well be finally emitted from the soil to the atmosphere in the form of CO2, if the dissolved organic carbon (DOC) is mineralized or if the soil water is lost via evapotranspiration, thus releasing the dissolved inorganic carbon (DIC)...This aspect of the importance of the downstream fate of leached carbon for the overall relevance of the leaching pathway for the carbon balance is missing in the manuscript.

**Response:**

This is a very good point! The downstream fate of the leached carbon is indeed very important as it will ultimately determine the relevance of leaching processes to the overall carbon budget or balance along the "soil-river-ocean" continuum. If part of the leached carbon is retained in the deeper soils or transformed and carried along from the river to the ocean in the form of DIC, it will not constitute a source of atmospheric $CO_2$ on a relatively short term (over years or decades). However, compared to DOC and DIC in the soil solution, the leached carbon is more likely to be subject to more intensified mineralization and outgassing during the land-ocean transfer, given more intensified mixing processes, oxygen exposure and photo-oxidation of terrestrial carbon upon releasing into the river (Hedges et al., 1997; Battin et al., 2009). Hence, we postulate that carbon leached from soils is more vulnerable to decomposition and/or release compared to that retained in the soil. That being said, it will be necessary to confirm our results and hypothesis using field-based leaching experiments to better understand the ultimate fate of leached soil carbon: whether it will be retained in the deeper soil or show a higher degradability upon leaving the soil matrix. Such information will be complementary to our study and further elucidate the importance of leaching processes in terms of ecosystem carbon budget.

The above considerations and discussions are now added as a separate paragraph in Section 3.4 in the revised paper:

> "An uncertainty related to the importance of leaching processes in the overall carbon budget along the "soil-river-ocean" continuum lies in the ultimate downstream fate of the leached carbon. If part of this carbon is retained in the surrounding soils or carried along from the river to the ocean in the form of DIC without outgassing into the air, it will not constitute a source of atmospheric $CO_2$ on a relatively short term (over years or decades). However, soil columns used in our study has a depth (60 cm) typical of or even deeper than the average soil depth in the alpine grasslands of Qinghai-Tibetan Plateau (Wang et al., 2001). Hence, we assume that carbon leached in our experiments will have minimum retention in the soil. Furthermore, compared to DOC and DIC in the soil solution, the leached carbon is more likely to be subject to more intensified mineralization and outgassing during the land-ocean transfer, given more intensified mixing processes, oxygen exposure and photo-oxidation of terrestrial carbon upon releasing into the river (Hedges et al., 1997; Battin et al., 2009). Hence, we postulate that

*carbon leached from soils is more vulnerable to decomposition and/or release compared to that retained in the soil. That being said, it will be necessary to confirm our results and hypothesis using field-based leaching experiments to better understand the ultimate fate of leached soil carbon: whether it will be retained in the deeper soil or show a higher degradability upon leaving the soil matrix. Such information will be complementary to our study and further elucidate the importance of leaching processes in terms of ecosystem carbon budget."*

References:

Battin, T. J., Luyssaert, S., Kaplan, L. A., Aufdenkampe, A. K., Richter, A., and Tranvik, L. J.: The boundless carbon cycle, Nature Geoscience, 2, 598-600, 2009.

Hedges, J. I., Keil, R. G., and Benner, R.: What happens to terrestrial organic matter in the ocean?, Org. Geochem., 27, 195-212, 1997.

Wang, S.Q., Zhu, S.L., and Zhou, C.H.: Characteristics of spatial variability of soil thickness in China, Geographical research, 20, 161-169, 2001.

**Comment 2:**

When judging the relevance of dissolved inorganic carbon leaching for the carbon balance, it is also crucial to differentiate between the biogenic fraction of DIC and its lithogenic (carbonate-rock derived) fraction. In my opinion, it is much more straight forward to compare the biogenic leaching losses of DIC with the NEP, than total DIC leaching losses. The authors might want to consider this in their discussion of their results in lines 25ff on page 9. In this context the application of the isotopic mass balance model is important. The results of this model depend strongly on the delta 13C values of the end-members carbonate and CO2 from soil respiration. The authors decided to use the delta 13C of the soil organic matter of -24 per-mille to calculate the biogenic fraction of dissolved inorganic carbon. Because isotopic fractionation occurs during the mineralization of soil organic matter, the authors might additionally use their delta 13C value of -23.1 per-mille as end member in order to assess the uncertainty that is associated with potential isotopic fractionation during mineralization and diffusive CO2 transport in soil (Cerling et al., 1991. On the isotopic composition of carbon in soil carbon dioxide. Geochim. Cosmochim. Acta 55, 3403-3405). Quantitatively more important than the isotopic fractionation during mineralization and diffusion of CO2 for the delta 13C value used as end member for the soil organic carbon derived fraction of DIC is the isotopic fractionation between CO2 in the gas phase and bicarbonate (Zhang et al., 1995. Carbon isotope fractionation during gas-water exchange and dissolution of CO2. Geochim. Cosmochim. Acta 59, 107-114). In the pH range of the investigated soils, the vast majority of the DIC will be present as bicarbonate (HCO3-). According to Zhang et al. (1995), isotope fractionation between the gas phase and the aqueous phase will shift the delta 13C of bicarbonate in equilibrium with gaseous CO2 by some 10-11 per-mille. Hence, the end member delta 13C of DIC in equilibrium with CO2, which has a delta 13C value of -24 per-mille, can be around -14 to -13 per-mille. Considering this isotopic fractionation between gaseous CO2 and bicarbonate will greatly increase the calculated fractions of biogenic (soil organic carbon-derived) DIC.

**Response:**

We must thank the reviewer for pointing out an excellent point that we overlooked. The dissolution of $CO_2$ produced by SOC degradation does cause large isotope fractionation on the biogenic carbonate/bicarbonate. Taking this into account, we have revised our endmember values ($\delta^{13}C_{biogenic-DIC}$ is estimated to be $-16‰$), and revised the following parts:

Section 2.4:

*"Similarly, the relative contribution of lithogenic carbonate and biogenic DIC derived from SOC degradation to leached DIC was assessed according to the following isotopic mass balance model:*

$$f_{carbonate} + f_{biogenic\text{-}DIC} = 1 \qquad (3)$$

$$f_{carbonate} \times \delta^{13}C_{carbonate} + f_{biogenic\text{-}DIC} \times \delta^{13}C_{biogenic\text{-}DIC} = \delta^{13}C_{DIC} \qquad (4)$$

*where $f_{carbonate}$ and $f_{biogenic\text{-}DIC}$ are proportion of carbonate- and biogenic DIC in total DIC; $\delta^{13}C_{carbonate}$ is the $\delta^{13}C$ value of soil carbonate, equivalent to 0‰ (Edwards and Saltzman, 2016); and $\delta^{13}C_{biogenic\text{-}DIC}$ is the $\delta^{13}C$ value of biogenic carbonate/bicarbonate derived from the dissolution of $CO_2$ produced by SOC degradation, which is estimated to shift by approximately 8‰ compared with the $\delta^{13}C$ value of soil-respired $CO_2$ (−24‰ here) due to isotope fractionation during $CO_2$ dissolution (Zhang et al., 1995). Hence, $\delta^{13}C_{biogenic\text{-}DIC}$ is estimated to be −16‰. $\delta^{13}C_{DIC}$ is the measured $\delta^{13}C$ signature of leached DIC. According to Hendy (1971) and Doctor et al. (2008), isotopic fractionation of leached DIC due to $CO_2$ loss in an open system is insignificant when the partial pressure of $CO_2$ ($pCO_2$) in the solution is lower than twice that of the surrounding atmosphere. Therefore, due to the much lower $pCO_2$ in the XLHT leachates (~ 200 μatm; Table S2) compared to that in the ambient atmosphere (> 400 μatm), the influence of $CO_2$ outgassing on the $\delta^{13}C$ of leached DIC was not considered in the present study."*

Section 3.3:

*"Based on the isotopic mass balance Eq. (3) and (4), lithogenic carbonate (with a $\delta^{13}C$ value of 0‰) contributed 51.2% to the leached DIC while biogenic DIC produced by SOC degradation contributed 48.4% (Fig. 5). The $\delta^{13}C$ value of leached DIC decreased to −12.3‰ and −13.5‰ during the second and third EPEs, corresponding to a contribution of 77.0% and 84.4% by biogenic sources in the total DIC, respectively (Fig. 5). ... Combined with the total flux rate, we calculated that both lithogenic and biogenic DIC fluxes were ~2.1 g C m$^{-2}$ in the first EPE. Subsequently, lithogenic DIC flux decreased to ~1.3 g C m$^{-2}$ while biogenic DIC flux increased to 7.6 g C m$^{-2}$ in the third EPE. This demonstrates that increased SOC degradation mainly contributed to the increased DIC fluxes with repeated EPEs."*

Section 3.4:

*"It is worth mentioning that biogenic DIC loss (16.0 ± 3.4 g C m$^{-2}$) caused by SOC degradation accounted for 184% of NEP at XLHT, indicating the importance of biogenic DIC to SIC loss during EPEs."*

**Comment 6:**

Starting on page 9, line 31, the authors argue that the carbon loss due to extreme precipitation events was much greater than carbon losses through warming-enhances respiration. This comparison is perhaps

misleading, because it implies that extreme precipitation events occur only as consequence of climate change. More correct would be the comparison of carbon losses due to warming-enhanced respiration with carbon losses due to "climate change-enhanced" extreme precipitation events.

**Response:**

The reviewer raised a good point. It is not fair to compare carbon leaching through all annual EPEs with warming-induced respiration increase. To be more consistent and robust, we decide to delete the discussion on the comparison of carbon losses due to warming-induced respiration with carbon losses due to annual EPEs.

**Response to Referee #2**

**General Comments:**

**Comment 1:**

The manuscript presents a study that attempt to evaluate the effect of extreme precipitation events on soil carbon losses in arid and semi-arid grasslands. The objective was to distinguish between C losses due to respiration and leaching. Additionally, leaching losses were separated into DIC and DOC losses. Therefore, a soil column experiment was conducted, were respiration and leaching losses were measured after an artificial precipitation events. Soil inorganic carbon losses due to leaching was higher than due to an enhanced respiration.

(1) As already mentioned by the first referee, the relevance of C losses depends on the fate of DIC and this should be more pronounced in the discussion.

(2) In addition, soil carbon losses due to DIC leaching has to be more discussed in detail, especially the fact that about 50% (or even more with the already recalculated values) of the DIC originates from SOC degradation (dissolved CO2). In consequence, the conclusion that most soil carbon during EPE is lost due to DIC (in particular SIC), might be not true. On the contrary, most of the DIC originates from dissolution of CO2, which originates from SOC mineralization and not SIC leaching. This should be discussed much more in detail.

**Response:**

(1) We have added one paragraph to discuss the fate of DIC in Section 3.4 as follows:

   *"An uncertainty related to the importance of leaching processes in the overall carbon budget along the "soil-river-ocean" continuum lies in the ultimate downstream fate of the leached carbon. If part of this carbon is retained in the surrounding soils or carried along from the river to the ocean in the form of DIC without outgassing into the air, it will not constitute a source of atmospheric $CO_2$ on a relatively short term (over years or decades). However, soil columns used in our study has a depth (60 cm) typical of or even deeper than the average soil depth in the alpine grasslands of Qinghai-Tibetan Plateau (Wang et al., 2001). Hence, we assume that carbon leached in our experiments will have minimum retention in the soil. Furthermore, compared to DOC and DIC in the soil solution, the leached carbon is more likely to be subject to more intensified mineralization and outgassing during the land-ocean transfer, given more intensified mixing processes, oxygen exposure and photo-oxidation of terrestrial carbon upon releasing into the river (Hedges et al., 1997; Battin et al., 2009). Hence, we postulate that carbon leached from soils is more vulnerable to decomposition and/or release compared to that retained in the soil. That being said, it will be necessary to confirm our results and hypothesis using field-based leaching experiments to better understand the ultimate fate of leached soil carbon: whether it will be retained in the deeper soil or show a higher degradability upon leaving the soil matrix. Such information will be complementary to our study and further elucidate the importance of leaching processes in terms of ecosystem carbon budget."*

(2) The reviewer raised an excellent point and we agree that the source of DIC needs to be discussed in order to determine the contribution of SIC dissolution to soil carbon loss. We have therefore calculated the flux of biogenic and lithogenic DIC from soils and added discussions in Sections 3.3 and 3.4 as follows:

   *"... Based on the isotopic mass balance of Eq. (3) and (4), lithogenic carbonate (with a $\delta^{13}C$ value of*

*0‰) contributed 51.2% to the leached DIC while biogenic DIC produced by SOC degradation contributed 48.4% (Fig. 5). The $\delta^{13}C$ value of leached DIC decreased to −12.3‰ and −13.5‰ during the second and third EPEs, corresponding to a contribution of 77.0% and 84.4% by biogenic sources in the total DIC, respectively (Fig. 5). These results confirm our previous hypothesis that SOC decomposition contributed significantly to soil DIC fluxes. Combined with the total flux rate, we calculated that both lithogenic and biogenic DIC fluxes were ~2.1 g C $m^{-2}$ in the first EPE. Subsequently, lithogenic DIC flux decreased to ~1.3 g C $m^{-2}$ while biogenic DIC flux increased to 7.6 g C $m^{-2}$ in the third EPE. This demonstrates that the increased DIC flux with repeated EPEs was mainly derived from increased contribution of SOC mineralization. Interestingly, increasing DIC fluxes with repeated EPEs were not observed in the KQ and GC soils (Fig. 4) despite their higher SOC contents and $CO_2$ release rates (Fig. S5). ..."*

*"Regardless of its source, the EPE-induced leaching loss of inorganic carbon was 31.5 and 10.6 μg DIC $g^{-1}$ soil from the alkaline XLHT and GC soils, respectively, approximately three and five times higher than the corresponding DOC leaching loss (5.9 and 3.9 μg DOC $g^{-1}$ soil, respectively). However, the KQ soil had a relatively lower EPE-induced DIC loss (4.4 μg DIC $g^{-1}$ soil) than the DOC leaching loss (11.6 μg DOC $g^{-1}$ soil) mainly due to its lower initial SIC content and relatively neutral soil pH value. Hence, DIC was the main form of soil carbon loss in alkaline soils during EPEs regardless of its source. When the source of the leached DIC is taken into account, dissolution of $CO_2$ produced by SOC mineralization (biogenic DIC) constituted more than half of the leached DIC (at least from the XLHT soils; Fig. 5), whose contribution increased with re-occurring EPEs (Fig. 5). This implies that SOC mineralization measured by $CO_2$ fluxes was under-estimated by approximately 8 times in the XLHT soils during the three EPEs (Fig. 5). In addition, DIC loss exclusively resulting from SIC dissolution or weathering was also a significant fraction of soil carbon loss, equivalent to 219% SOC loss in the form of $CO_2$ during EPEs (Fig. 5). These results collectively corroborate that inorganic carbon loss is the main form of soil carbon loss in alkaline soils during EPEs."*

*"... In view of DIC sources, biogenic DIC loss derived from SOC mineralization contributed to more than half of the total leached DIC fluxes and accounted for 184% of the NEP at XLHT. ... These results also imply that SOC mineralization measured by $CO_2$ fluxes might be drastically under-estimated in alkaline grassland soils during EPEs. ..."*

**Comment 2:**

The experimental setup seems appropriate for the objectives presented in the manuscript, however the presented results need some reconsideration and recalculation, especially respiration data should be presented as specific respiration to account for different SOC contents in the investigated soils (for more detail see specific comments).

**Response:**

Following the reviewer's suggestion, we calculate the specific respiration data normalization to SOC and added in Sections 3.2 and Figure 3b as follows:

*"... The specific soil respiration rates normalized to SOC were 2.2, 2.6, and 7.9 μg C $g^{-1}$ SOC $h^{-1}$ in the GC, KQ, and XLHT soils, respectively. This indicated that SOC in the XLHT soils was easier to degrade despite*

*its low content."*

Figure:

[Figure]

*"Figure 3: Total (a) and specific (b) extreme precipitation event (EPE)-induced $CO_2$ release in the litter-amended and non-amended grassland soils during three EPEs. Mean values are shown with standard deviation (n = 3). Lower-case letters (a, b, c) indicate significantly different levels among the litter-amended and non-amended soils determined by Duncan's multiple range test (one-way ANOVA, $p < 0.05$)."*

Section 3.2:

*"Using data shown in Fig. S3-4, we calculated that total EPE-induced $CO_2$ release during three EPEs was higher in the KQ and GC soils than in the XLHT soil ($p < 0.05$; Fig. 3a) with a lower SOC content and a lower SOC:N ratio (Table 1). However, the specific EPE-induced $CO_2$ release normalized to SOC content showed no significant difference in the non-amended soils among three sites (Fig. 3b), indicating that a similar proportion of SOC (~4%) was subject to EPE-induced $CO_2$ release in the alpine and temperate grassland soils. The total EPE-induced $CO_2$ release (including $CO_2$ from both litter and SOC mineralization) was significantly higher in the litter-amended KQ soils than the non-amended ones, similar to the specific EPE-induced $CO_2$ in the KQ and XLHT soils. The specific EPE-induced $CO_2$ was significantly different for the litter-amended soils among sites ($p < 0.05$), showing a pattern of KQ > XLHT > GC. This pattern was consistent with the mean sand content in the order of KQ (46.9%) > XLHT (39.1%) > GC (27.2%). In addition, the higher total and specific EPE-induced $CO_2$ release in the litter-amended KQ soils may be related to its relatively lower soil pH (~7.7), which facilitates the release rather than the dissolution of respired $CO_2$ in soil solution. We therefore conclude that the KQ soil, with a coarser texture and a lower pH (Table 1), may have provided less sorptive protection for the labile DOC components after EPEs (Kell et al., 1994; Nelson et al., 1994) and allowed less dissolution of the respired $CO_2$, and hence showed a more responsive respiration to the precipitation events. These results suggest that SOC contents and SOC:N ratios are important factors influencing the total EPE-induced $CO_2$ release, while the availability of labile*

*organic carbon, soil texture and pH are key factors affecting the specific EPE-induced $CO_2$ release in these grassland soils.*"

**Specific Comments:**

**Comment 3:**

Page 3, line 25: The dimension of the soil pits seems quite small 10 cm x 10 cm. Even by using a shovel I would expect that you need a bigger area to go down to 70 cm.

**Response:**

Thank you for pointing out this inaccurate description. We have confirmed that the pit was 25 cm × 25 cm × 70 cm. This is corrected in the text.

**Comment 4:**

Page 5, line 1: How was the water added to the soil columns? Did you have needles in the top lid of the soil column? Did you used a constant rate, like 0.5 mm per minute? How much time was in between the EPE events? How long did you wait until you start a new EPE? How where the soil columns treated in between the CO2 measurements? Were they closed or flushed with constant air flow? Please provide more information about the experimental setup for the reader.

**Response:**

More information about the experimental setup is added at Section 2.2 as follows:

"*...approximately 1 L of rainwater (rainfall of ~127 mm), comparable to 30% of the MAP of the investigated sites, was added to the surface of each soil column over 3–4 h at rates of one drop per second using syringes and allowed to leach through the column to be collected with a clean beaker within 12–14 h. ... To monitor soil respiration every 1–2 days following each EPE, soil columns were first aerated for 1 h using $CO_2$-depleted air that had been passed through saturated sodium hydroxide (NaOH) solutions (twice; Fig. 1) and then incubated for 4 h with lids closed. $CO_2$ gas in the column headspace was collected by gas-tight syringes for the subsequent measurement. After collection of $CO_2$ gas, the lids were open to allow the exchange with the ambient air. Soil respiration was monitored for 30 days after the first EPE and observed to stabilize approximately on the 20th day (Fig. S3). Hence, the first, second, and third EPEs were conducted on the 1st, 31st, and 51st day of incubation, and the $CO_2$ measurement was conducted for approximately 30, 20, and 20 days after the first, second, and third EPEs, respectively.*"

**Comment 5:**

Page 7, line 8: Additionally, to the respiration rate I'd suggest to calculate a specific respiration rate, which is the respiration rate divided by the amount of SOC (mg CO2-C g-1 SOC h-1). This would allow an easier comparison of the different soils with different SOC content.

**Response:**

The specific respiration rate has been added in Section 3.2 as follows:

"*...The specific soil respiration rates normalized to SOC were 2.2, 2.6, and 7.9 $\mu g\, C\, g^{-1}\, SOC\, h^{-1}$ in the GC, KQ, and XLHT soils, respectively. This indicated that SOC in the XLHT soils was easier to degrade despite*

*its low content. "*

**Comment 6:**

Page 7 line 13: The authors argue that litter addition increase respiration due to mineralization of labile litter compounds and priming effects. However, here were no values presented which would underline this statement. Since the authors measured $\delta^{13}CO_2$, they should be able to separate litter mineralization from SOC mineralization.

**Response:**

Good point! We take the GC soils as an example to quantify the relative contribution of litter and SOC mineralization to the respired $CO_2$ in the following parts and added one inside-figure in Fig. S3:

Figure:

[Figure]

Inside figure of Fig. S3: Variations of litter-derived and SOC-derived $CO_2$ release with time in the litter-amended GC soils during the first EPE.

Section 3.2

*"Total respired $CO_2$ was higher in the litter-amended than non-amended soils before and after EPEs (Fig. S3), likely caused by one or two following reasons: (1) the degradation of labile components in the fresh litter; (2) increased degradation of SOC primed by litter additions (Fröberg et al., 2005; Ahmad et al., 2013). These two reasons might affect $CO_2$ release in litter-amended soils in two ways: (1) both of them had positive effects; (2) one of them had positive effects and the other one had negative ones, but the total net effects was positive. To distinguish the influences of above two reasons on total respired $CO_2$ and further differentiate the contribution of litter (C4) and SOC (C3) to the respired $CO_2$, we examined the $\delta^{13}C$ values of $CO_2$ evolved from the GC soils after the first EPE. On the first day after EPE, $CO_2$ from the non-amended and litter-amended GC soils had a $\delta^{13}C$ value of –23.1‰ and –18.7‰, respectively. The latter was close to the $\delta^{13}C$ signature of the added litter (–16.25‰). Using the two-endmember mixing model of Eq. (1) and (2), we calculated that litter contributed ~64% of the respired $CO_2$ in the litter-amended GC soils. However, along with the consumption of labile carbon in litter, the $\delta^{13}C$ signature of $CO_2$ decreased from –18.7‰ on*

*Day 1 to –21.8‰ on Day 25 after EPE in the litter-amended soils (Fig. 2). Accordingly, the proportion of litter-derived $CO_2$ decreased from 64% to 20%. The litter-derived $CO_2$ flux in litter-amended GC soils was estimated to range from 7.0 to 17.5 mg C m$^{-2}$ h$^{-1}$, while the SOC-derived $CO_2$ flux increased from 6.2 to 15.7 mg C m$^{-2}$ h$^{-1}$ after the first EPE (Fig. S3). Compared with the SOC-derived $CO_2$ flux in non-amended GC soils (ranging from 17.2 to 27.1 mg C m$^{-2}$ h$^{-1}$), litter addition had a negative priming effect on the degradation of native SOC while increasing total respiration through labile litter degradation."*

**Comment 7:**

Page 7, line 17: (1) The used mixing model should be mentioned in the method section. (2) Further with the given isotopic values I cannot understand how the authors calculated contribution of litter mineralization to total respiration. Using a mixing model of: 1 – ((c_mix-c_litter)/(c_control-c_litter)), where c_mix is the isotopic value of CO2 from the litter amended sample (-18,7‰, c_litter the isotopic value of the added litter (-16,2‰ and c_control the isotopic value of CO2 from the non-amended sample (-23,1‰, the contribution of litter mineralization to total respiration was around 64% at day 1 and only 19% at day 25. Which values for δ13CO2 values did you used for the control (non- amended) samples. Did you measured δ13CO2 for the control only at the beginning or at the same resolution as δ13CO2 for the litter-amended samples? (3) Further, are there any isotopic measurements of the other two sites. If so, why they are not shown here?

**Response:**

(1) The description on the mixing model has been added in Section 2.4 as follows:

*"The relative contribution of litter- and SOC-derived $CO_2$ to total respired $CO_2$ in the litter-amended soils was estimated using the following mass balance model:*

$$f_{litter\text{-}derived} + f_{SOC\text{-}derived} = 1 \qquad (1)$$

$$f_{litter\text{-}derived} \times \delta^{13}C_{litter\text{-}derived} + f_{SOC\text{-}derived} \times \delta^{13}C_{SOC\text{-}derived} = \delta^{13}C_{respired\text{-}CO2} \qquad (2)$$

*where $f_{litter\text{-}derived}$ and $f_{SOC\text{-}derived}$ are the proportion of litter- and SOC-derived $CO_2$ in the total respired $CO_2$; $\delta^{13}C_{litter\text{-}derived}$ is the $\delta^{13}C$ value of litter-derived $CO_2$, equivalent to $-16.25‰$; $\delta^{13}C_{SOC\text{-}derived}$ is the $\delta^{13}C$ value of SOC-derived $CO_2$, which assumes the same value as that in the non-amended soils at the beginning of incubation (–23.1‰) according to Cerling et al. (1991); $\delta^{13}C_{respired\text{-}CO2}$ is the measured $\delta^{13}C$ of respired $CO_2$."*

(2) Regarding the calculation results on the proportion of litter-derived $CO_2$, I must admit that I had made a very stupid mistake. I previously wrongly calculated the value of (c_control–c_litter) as −8.95‰, which actually was −6.85‰ (i.e., (−23.1‰) − (−16.25‰)), and this mistake resulted in the incorrect proportions (72% to 39%). I have revised these values and re-checked all the data in this manuscript. The revised parts are as follows:

*"Using the two-endmember mixing model of Eq. (1) and (2), we calculated that litter contributed ~64% of the respired $CO_2$ in the litter-amended GC soils. However, along with the consumption of labile carbon in litter, the $\delta^{13}C$ signature of $CO_2$ decreased from –18.7‰ on Day 1 to –21.8‰ on Day 25 after EPE in the litter-amended soils (Fig. 2). Accordingly, the proportion of litter-derived $CO_2$ decreased from 64% to 20%."*

(3) Due to budget constraints and logistic reasons, we measured the isotopes of respired $CO_2$ only in the GC soil during the first extreme precipitation event. The other two soils were not measured. As the present study mainly aimed to compare the relative importance of respiration and leaching in EPE-

induced soil carbon loss from grassland soils, the isotopic analysis was conducted for only one soil to provide some information regarding the source(s) of the leached and respired carbon.

**Comment 8:**

Page 7 line21.: (1) Despite the fact that the calculation described here might be simple, it should be part of the method section and not of the result/discussion section. (2) "EPE-induced CO2 release was higher in the KQ and GC soils than in the XLHT soil ($p < 0.05$; Fig. 3) that had a lower SOC content and a lower SOC:N ratio (Table 1)", as mentioned above, I suggest to calculate a specific respiration normalized to the absolute amount of SOC in the soil column. The specific respiration will provide more information about the stability and the loss of C from the different sites. A rough calculation based on figure 3 revealed that respiration of the 3 sites in the non-amended treatment might not differ. However, this has to be checked with the measurement values. It is also not clear to which EPE is shown in Fig 3, is it the first, second or third one?

**Response:**

(1) Calculation on EPE-induced $CO_2$ release has been moved to the method Section 2.4 as follows:
   *"EPE-induced $CO_2$ release via respiration was assessed following two steps. First, cumulative respiration during the first 20 days after each EPE (until respiration rate stabilized) was calculated. Second, difference between the measured cumulative respiration and that estimated using the stabilized basal respiration rate after each EPE was calculated as the EPE-induced $CO_2$ release."*
(2) Results on the specific EPE-induced $CO_2$ release normalized to the SOC content has been added as Fig. 3b, and discussed in Section 3.2 (please see our response to Comment 2).

**Comment 9:**

Page 7, line 25: "Litter amendment significantly increased the EPE-induced CO2 release from the KQ soil ($p < 0.05$) but did not have any effect on the XLHT and GC soils." - I disagree with this statement, because figure 3 only show total CO2 release which is the sum of CO2 from litter mineralization and SOC mineralization from the mineral soil. Therefore, it is only possible to say that EPE increased total CO2 release. However, as mentioned above, I would recommend to use the isotopic data of CO2 to separate respiration into the two sources litter and soil. Which would allow a comparison of SOC mineralization between non-amended and amended soils.

Further the authors explained the higher CO2 release in the KQ soil with a coarser soil texture and less sorptive protection of DOC, however I can't follow this explanation since Table 1 show a lower sand content in the KQ than in the XLHT soils. "These results suggest that soil texture, SOC content and quality are important factors influencing EPE-enhanced soil respiration." This is a quite strong statement, because there is no information given about the SOC quality in the different soils used in this study, also the soil texture of the used soils is more or less similar and the presented results does not justify such a strong statement. In fact, the higher CO2 release in the KQ can be explained by several other factors, such as the higher pH of the GC and XLHT soils. The higher the pH the more CO2 is dissolved, which can be seen in Fig 4 in the higher DIC contents of these sites. Also negative priming effects may occur in the GC and XLHT sites which lead to a reduced SOC mineralization, because microorganisms utilize the added litter first. This should be discussed in more detail.

**Response:**

We agree with the first part of the comment and have quantified the litter- and SOC-derived $CO_2$ as details below. As mentioned in our replies to previous comments, the isotopes of respired $CO_2$ were only measured in the GC soils during the first extreme precipitation event due to logistic reasons. Nonetheless, we have added detailed discussion on the total EPE-induced $CO_2$ release as well as its influencing factors in the following parts:

Section 3.2

*"Total respired $CO_2$ was higher in the litter-amended than non-amended soils before and after EPEs (Fig. S3), likely caused by one or two following reasons: (1) the degradation of labile components in the fresh litter; (2) increased degradation of SOC primed by litter additions (Fröberg et al., 2005; Ahmad et al., 2013). These two reasons might affect $CO_2$ release in litter-amended soils in two ways: (1) both of them had positive effects; (2) one of them had positive effects and the other one had negative ones, but the total net effects was positive. To distinguish the influences of above two reasons on total respired $CO_2$ and further differentiate the contribution of litter (C4) and SOC (C3) to the respired $CO_2$, we examined the $\delta^{13}C$ values of $CO_2$ evolved from the GC soils after the first EPE. On the first day after EPE, $CO_2$ from the non-amended and litter-amended GC soils had a $\delta^{13}C$ value of –23.1‰ and –18.7‰, respectively. The latter was close to the $\delta^{13}C$ signature of the added litter (–16.25‰). Using the two-endmember mixing model of Eq. (1) and (2), we calculated that litter contributed ~64% of the respired $CO_2$ in the litter-amended GC soils. However, along with the consumption of labile carbon in litter, the $\delta^{13}C$ signature of $CO_2$ decreased from –18.7‰ on Day 1 to –21.8‰ on Day 25 after EPE in the litter-amended soils (Fig. 2). Accordingly, the proportion of litter-derived $CO_2$ decreased from 64% to 20%. The litter-derived $CO_2$ flux in litter-amended GC soils was estimated to range from 7.0 to 17.5 mg C $m^{-2}$ $h^{-1}$, while the SOC-derived $CO_2$ flux increased from 6.2 to 15.7 mg C $m^{-2}$ $h^{-1}$ after the first EPE (Fig. S3). Compared with the SOC-derived $CO_2$ flux in non-amended GC soils (ranging from 17.2 to 27.1 mg C $m^{-2}$ $h^{-1}$), litter addition had a negative priming effect on the degradation of native SOC while increasing total respiration through labile litter degradation."*

*"Using data shown in Fig. S3-4, we calculated that total EPE-induced $CO_2$ release during three EPEs was higher in the KQ and GC soils than in the XLHT soil (p < 0.05; Fig. 3a) with a lower SOC content and a lower SOC:N ratio (Table 1). However, the specific EPE-induced $CO_2$ release normalized to SOC content showed no significant difference in the non-amended soils among three sites (Fig. 3b), indicating that a similar proportion of SOC (~4%) was subject to EPE-induced $CO_2$ release in the alpine and temperate grassland soils. The total EPE-induced $CO_2$ release (including $CO_2$ from both litter and SOC mineralization) was significantly higher in the litter-amended KQ soils than the non-amended ones, similar to the specific EPE-induced $CO_2$ in the KQ and XLHT soils. The specific EPE-induced $CO_2$ was significantly different for the litter-amended soils among sites (p < 0.05), showing a pattern of KQ > XLHT > GC. This pattern was consistent with the mean sand content in the order of KQ (46.9%) > XLHT (39.1%) > GC (27.2%). In addition, the higher total and specific EPE-induced $CO_2$ release in the litter-amended KQ soils may be related to its relatively lower soil pH (~7.7), which facilitates the release rather than the dissolution of respired $CO_2$ in soil solution. We therefore conclude that the KQ soil, with a coarser texture and a lower pH (Table 1), may have provided less sorptive protection for the labile DOC components after EPEs (Kell et al., 1994; Nelson et al., 1994) and allowed less dissolution of the respired $CO_2$, and hence showed a more responsive respiration to the precipitation events. These results suggest that SOC contents and SOC:N ratios are important factors influencing the total EPE-induced $CO_2$ release, while the availability of labile organic carbon, soil texture and pH are key factors affecting the specific EPE-induced $CO_2$ release in these grassland soils."*

**Comment 10:**

Page 8 line 8: The authors mentioned that CO2 from litter decomposition contributed to DIC, however in the following lines the argue that this effect is more pronounced in the KQ soils than in the XLHT and GC, due to the higher SIC content of the later once. This is true speaking in relative amounts, but considering absolute amounts I would expect that much more CO2 is dissolved in the GC and XLHT soils due to the higher pH Therefore CO2 might also be a source of DIC in these soils with high SIC content. Which is also shown in line 23-24 on the same page.

**Response:**

The relative effect of litter amendment on DIC fluxes is now discussed in this paragraph. Because the added litter-OC was only 0.7 g per column calculated using added litter biomass of 1.59 g and litter-OC content of 43%, and the maximum DIC content derived from dissolution of $CO_2$ produced by litter degradation was 0.8 g assuming that all the litter-OC was mineralized and that all the produced $CO_2$ was dissolved in soil solution. Compared with the SIC content (38.15 g C per column) in XLHT soils, litter-mineralization-derived DIC was far smaller, but the dissolved CO2 derived from litter-OC mineralization could be more important in the KQ soils having only 0.7 g SIC per column. To clarify this, we have revised this paragraph as follows:

*"... Due to the high SIC content in the XLHT soils (38.15 g per column) and the low litter-OC amendment (0.7 g per column), there was no significant difference of DIC fluxes between the non-amended and litter-amended XLHT soils (Fig. 4a). However, for the KQ soil having a relatively low SIC content which was similar to the added litter-OC (0.7 g per column; Table 1), the influence of litter addition on the DIC flux was quite obvious. Therefore, although the contribution of dissolved $CO_2$ to DIC fluxes should be more important in high-pH soils, the relative effect of litter amendment on DIC fluxes under EPEs seemed more significant in soils with a low SIC content."*

**Comment 11:**

Page 8, line 19f.: Why was this measurement only done for the XLHT soils? Or why the results from the other sites not shown?

**Response:**

Due to budget constraints and logistic reasons, we only measured the isotopes of leached DIC in the XLHT soils and took it as an example to trace the source of the leached DIC. This is now mentioned in the Methods.

**Comment 12:**

Page 8, line 27f.: "Interestingly, increasing DIC fluxes were not observed in the KQ and GC soils (Fig. 4), although they had higher SOC content and degradation (i.e., respiration) rates (Fig. S5)" - What do you mean with higher degradation?

**Response:**

Given the above comments and discussions, it may be inaccurate to treat the degradation rate as the same as the $CO_2$ release rate. To clarify, we have revised this sentence as follows:

*"Interestingly, increasing DIC fluxes with repeated EPEs were not observed in the KQ and GC soils (Fig. 4) despite their higher SOC contents and $CO_2$ release rates (Fig. S5)."*

**Comment 13:**

Page 8, line 31: Here the authors calculated a specific leaching loss normalized to SIC content and compared it to DOC losses. However, in line 9 same page, it is mentioned that especially for the soils with low SIC content, the dissolution $CO2$ from SOC and litter mineralization is a major source for DIC. Therefore, the high value of 32.5 mg C g-1 SIC has to attributed to SOC mineralization and not to SIC loss. In consequence, the drawn conclusion that SIC loss is the main form of soil carbon loss in neutral to alkaline soils might be questioned, at least for the KQ soils.

**Response:**

Good point! It is inaccurate to say that SIC loss is the main form of soil carbon loss in neutral to alkaline soils. We have now used DIC flux rates in the units of mg C $g^{-1}$ soil to compare with SOC loss. In addition, regardless of the DIC sources, DIC (not SIC) was a major form of carbon loss from these grassland soils. In view of its sources, biogenic DIC should not be neglected. Therefore, we have revised Figure 5 and added one paragraph in Section 3.3 to discuss the role of biogenic DIC in leached DIC.

Figure :

[Figure]

*"Figure 5: The flux of carbon loss from soil organic carbon (SOC) mineralization including $CO_2$ release and biogenic dissolved inorganic carbon (DIC), and that from soil carbon dissolution including leached dissolved organic carbon (DOC) and lithogenic DIC in the XLHT soils. Mean values are shown with standard error (n = 3)."*

Section 3.3

*"Regardless of its source, the EPE-induced leaching loss of inorganic carbon was 31.5 and 10.6 µg DIC $g^{-1}$ soil from the alkaline XLHT and GC soils, respectively, approximately three and five times higher than the corresponding DOC leaching loss (5.9 and 3.9 µg DOC $g^{-1}$ soil, respectively). However, the KQ soil had a relatively lower EPE-induced DIC loss (4.4 µg DIC $g^{-1}$ soil) than the DOC leaching loss (11.6 µg DOC $g^{-1}$ soil) mainly due to its lower initial SIC content and relatively neutral soil pH value. Hence, DIC was the main form of soil carbon loss in alkaline soils during EPEs regardless of its source. When the source of the*

*leached DIC is taken into account, dissolution of $CO_2$ produced by SOC mineralization (biogenic DIC) constituted more than half of the leached DIC (at least from the XLHT soils; Fig. 5), whose contribution increased with re-occurring EPEs (Fig. 5). This implies that SOC mineralization measured by $CO_2$ fluxes was under-estimated by approximately 8 times in the XLHT soils during the three EPEs (Fig. 5). In addition, DIC loss exclusively resulting from SIC dissolution or weathering was also a significant fraction of soil carbon loss, equivalent to 219% SOC loss in the form of $CO_2$ during EPEs (Fig. 5). These results collectively corroborate that inorganic carbon loss is the main form of soil carbon loss in alkaline soils during EPEs."*

**Comment 14:**

Page 9, line 33: "It is also worth mentioning that soil carbon leaching fluxes in this study ($10.1-25.3$ g C $m^{-2}$ $yr^{-1}$) far exceed annual SOC loss through warming-enhanced respiration at these sites ($0.2-0.6$ g C $m^{-2}$ $yr^{-1}$) given an assumed temperature sensitivity of 2 in climate..." - I think the authors can not make such a comparison, since most of the DIC in this study originates from SOC degradation, therefore DIC losses has to be attributed to respiration losses and not DIC losses. Meaning that dissolution of $CO_2$ needs first mineralization of SOC by microorganism.

**Response:**

After considering both Reviewer #1 and #2's comments, we have removed this part to make this manuscript more rigorous and accurate.

**Comment 15:**

Technical corrections Fig. S4 and S5 are swapped, the caption of Fig. S4 belongs to Fig. S5 and vice versa.

**Response:**

Revised. Thank you!

[revised manuscript text omitted]

**Contents of this file**

Table S1 to S2

Figure S1 to S5

**Additional Supporting Information (Files upload separately)**

Dataset S1

**Introduction**

The supplementary material contains Table S1-S2, Figures S1-S5, and Dataset S1. Table S1 presents the major ion concentrations of the artificial rainwater used for the simulated extreme precipitation events (EPEs). Table S2 shows proportion of outgassed $CO_2$ in total dissolved inorganic carbon under given pH, temperature and alkaline conditions. Fig. S1 provides the locations of sample sites and their mean annual precipitation (MAP) levels. Fig. S2 provides daily and monthly precipitation in Xilinhot. Fig. S3 shows soil respiration rate variation after EPEs. Fig. S4 provides information on soil carbon content in each column. Fig. S5 provides information on cumulative repiration in soils of different treatments. Dataset S1 (separate file) shows data displayed in the figures of main text and supporting information.

**Table S1: Major ion concentrations (µmol L$^{-1}$) of the artificial rainwater used for the simulated extreme precipitation events (EPEs).**

| Ion | For XLHT and KQ soils | For GC soils |
|---|---|---|
| $Ca^{2+}$ | 113 | 180 |
| $Mg^{2+}$ | 17 | 37 |
| $Na^+ + K^+$ | 41 | 87 |
| $HCO_3^-$ | 180 | 12 |
| $Cl^-$ | 45 | 75 |
| $SO_4^{2-}$ | 161 | 67 |

**Table S2: Proportion of outgassing CO2 in total dissolved inorganic carbon under given pH, temperature and alkaline conditions according to formulas shown in Ran et al. (2015).**

| pH[a] | T[b] | Alkaline [c] (μmol/l) | Proportion of outgassing $CO_2$ (%) | Partial Pressure of $CO_2$ (μatm) |
|---|---|---|---|---|
| 9.0 | 23 | 300 | 0.2 | 18 |
| 8.5 | 23 | 300 | 0.7 | 59 |
| 7.5 | 23 | 300 | 6.8 | 601 |
| 9.0 | 23 | 500 | 0.2 | 29 |
| 8.5 | 23 | 500 | 0.7 | 98 |
| 7.5 | 23 | 500 | 6.8 | 1002 |
| 9.0 | 23 | 1000 | 0.2 | 58 |
| 8.5 | 23 | 1000 | 0.7 | 195 |
| 7.5 | 23 | 1000 | 6.8 | 2004 |
| 9.0 | 23 | 2000 | 0.2 | 117 |
| 8.5 | 23 | 2000 | 0.7 | 391 |
| 7.5 | 23 | 2000 | 6.8 | 4008 |
| 9.0 | 23 | 3000 | 0.2 | 175 |
| 8.5 | 23 | 3000 | 0.7 | 586 |
| 7.5 | 23 | 3000 | 6.8 | 6013 |

[a]: pH of the soil leachates in this study, $pH_{XLHT} \sim 9$, $pH_{KQ} \sim 7.6$, $pH_{GC} \sim 8.6$;

[b]: incubation temperature;

[c]: alkaline range of natural waters.

[Figure]

**Figure S1: Map of sampling sites and their mean annual precipitation (MAP) levels.**

[Figure]

**Figure S2: Variation of the daily precipitation (a) and monthly precipitation (b) during recent two decades at Xilinhot in Inner Mongolia (data modified from http://data.cma.cn/data/index/6d1b5efbdcbf9a58.html).**

[Figure]

**Figure S3: Variations of soil CO₂ release rate after the simulated extreme precipitation events (EPEs). The inside-figure shows the variations of litter-derived and SOC-derived CO₂ release with time in litter-amended GC soils during the first EPE. Red and blue dots represent respiration in the litter-amended and non-amended soils, respectively. Mean values are shown with standard error (n = 3). Red and blue dash lines represent the stabilized respiration rate after EPEs in the litter-amended and non-amended soils, respectively. Black dash lines represent regressive lines using polynomial fitting method.**

[Figure]

**Figure S4: Total amount of soil organic carbon (SOC) and soil inorganic carbon (SIC) in different layers of the artificial soil columns.**

[Figure]

**Figure S5: Cumulative respiration in the non-amended and litter-amended soils during the first 20 days after three extreme precipitation events (EPEs) relative to that estimated using basal respiration rates stabilized after EPEs. Mean values are shown with standard error (n = 3).**

---

## Referee Report (RR1)

[referee-annotated manuscript omitted]

---

## Author Response (AR2)

Dear Editor,

Thank you for allowing us to submit a revised version of our manuscript. We greatly appreciate the detailed and constructive comments of you and two reviewers which helped us to improve the manuscript. We addressed all comments as attached below (highlighted in the manuscript in blue for reviewer 1's comments and in red for reviewer 2's comments) and hope that we adequately solved the requests.

With kind regards,

Xiaojuan Feng

(We attached a marked version of the manuscript and the supplement with changes highlighted at the end of this pdf.)

**Response to Editor**

**Comment:**

Both reviewers agree that your manuscript has improved substantially, but several issues remain. Reviewer 1 provides some very useful editorial suggestions; reviewer 2 provides important editorial suggestions as well.

In addition, reviewer 2 points out a few results that really need a closer look; for instance, I fully agree with reviewer 2 that the reported specific soil respiration rates on page 7 are not consistent with reported soil C contents and total respiration rates (compared to the other two soils, the XLHT soil has roughly 60% less organic carbon, yet the respiration rates are also about 60% lower. So, the specific respiration rate should not be substantially higher than for the other soils). Reviewer 2 also suggests to include a brief discussion on two issues, and I agree on both accounts. In your revised version, please address all comments in detail.

**Response:**

Thank you. Below we provide a point-to-point response to all the issues raised by the referee. We hope our replies and revisions will satisfy all the requests.

**Response to Referee #1**

Thank you for providing so much detailed suggestions on improving this manuscript. All the comments are very constructive and helpful, and we have revised the manuscript thoroughly. Suggestions including language correction and rearrangements of sentences have been adopted accordingly, and we supply a pdf showing the point-to-point response. Here we only demonstrate the response to comments that need to discuss in detail.

**Comment 1:**

"It should be mentioned that the DIC concentration may be underestimated due to $CO_2$ outgassing during leachate collection. However, the potential underestimation is lower than 7% owing to the low proportion of outgassed $CO_2$ in total DIC (Table S2) as calculated using formulas in Ran et al. (2015)."
Table S2 indicates that there is outgassing of $CO_2$ from solution into the atmosphere even, for $pCO_2$ in solution that are smaller than the atmospheric $pCO_2$. How can this be?

**Response:**

First of all, we apologize for the incorrect use of the term "outgassing". We actually meant "exchange". According to Table S2, the calculated partial pressure of $CO_2$ in the leachate ranged from 18–409, 59–1368 and 601–14029 µatm in the XLHT, KQ, and GC soils, respectively, under a series of alkalinity conditions (300–7000 µmol/L) using our leachate pH (7.5–9.0) and incubation temperature (23 °C) according to equations listed in Ran et al. (2015). Hence, depending on the assumed alkalinity, leachate $pCO_2$ could be higher or lower than the mean atmospheric $pCO_2$ (400–500 µatm in our lab) and $CO_2$ exchanges in both directions (outgassing and absorption) could happen. However, as the leachate pH was high (7.5–9.0) in all samples, bicarbonate ($HCO_3^-$) rather than dissolved $CO_2$ (more abundant in acidic conditions) was the predominant form in the leachates. We estimate that dissolved $CO_2$ represented <7% (0.2–6.8%) of total DIC ($HCO_3^-$ + $CO_3^{2-}$ + dissolved $CO_2$) in the range of pH we measured. Therefore, regardless of outgassing or absorption, exchange-induced alterations in DIC concentrations were relatively small. We have revised this sentence and Table S2 as follows:

*"It should be mentioned that the DIC concentration may vary due to exchanges between dissolved and atmospheric $CO_2$ during leachate collection. However, potential contribution from this process was < 7% owing to the low proportions of dissolved $CO_2$ in total DIC of our samples (Table S2) as calculated using formula in Ran et al. (2015)."*

**Table S2**: *Proportion of dissolved $CO_2$ in total dissolved inorganic carbon under the given pH, temperature and alkalinity conditions.*

| pH[a] | T[b] | Alkalinity[c] (µmol/L) | Proportion of dissolved[d] $CO_2$ (%) | Partial Pressure of $CO_2$[d] (µatm) |
|---|---|---|---|---|
| 9.0 | 23 | 300 | 0.2 | 18 |
| 8.5 | 23 | 300 | 0.7 | 59 |
| 7.5 | 23 | 300 | 6.8 | 601 |
| 9.0 | 23 | 500 | 0.2 | 29 |
| 8.5 | 23 | 500 | 0.7 | 98 |
| 7.5 | 23 | 500 | 6.8 | 1002 |
| 9.0 | 23 | 1000 | 0.2 | 58 |
| 8.5 | 23 | 1000 | 0.7 | 195 |
| 7.5 | 23 | 1000 | 6.8 | 2004 |
| 9.0 | 23 | 2000 | 0.2 | 117 |
| 8.5 | 23 | 2000 | 0.7 | 391 |
| 7.5 | 23 | 2000 | 6.8 | 4008 |
| 9.0 | 23 | 3000 | 0.2 | 175 |
| 8.5 | 23 | 3000 | 0.7 | 586 |

| 7.5 | 23 | 3000 | 6.8 | 6013 |
|-----|----|------|-----|------|
| 9.0 | 23 | 7000 | 0.2 | 409 |
| 8.5 | 23 | 7000 | 0.7 | 1368 |
| 7.5 | 23 | 7000 | 6.8 | 14029 |

[a]: pH of the soil leachates in this study, $pH_{XLHT} \sim 9$, $pH_{KQ} \sim 7.6$, $pH_{GC} \sim 8.6$;

[b]: incubation temperature;

[c]: alkalinity range of natural and soil waters.

[d]: calculation of partial pressure and proportion of dissolved $CO_2$ was based on equations in Ran et al. (2015).

**Comment 2:**

"According to Hendy (1971) and Doctor et al. (2008), isotopic fractionation of leached DIC due to $CO_2$ loss in an open system is insignificant when the partial pressure of $CO_2$ ($pCO_2$) in the solution is lower than twice that of the surrounding atmosphere. Therefore, due to the much lower $pCO_2$ in the XLHT leachates (~ 200 µatm; Table S2) compared to that in the ambient atmosphere (> 400 µatm), the influence of $CO_2$ outgassing on the $\delta^{13}C$ of leached DIC was not considered in the present study."
How can the $pCO_2$ that was established in soil water in contact with soil air be smaller than the $pCO_2$ of atmospheric air? The $pCO_2$ in soil air is typically higher than the atmospheric $pCO_2$.

**Response:**

Again, according to Table S2, we calculated $pCO_2$ in the soil leachates under a series of alkalinity conditions (300–7000 µmol/L) using the measured leachate pH and incubation temperature (23 °C). Leachates from the XLHT soil had a lower dissolved $CO_2$ concentration and hence lower $pCO_2$ (18–409 µatm) compared to other two soils (59–1368 and 601–14029 µatm in the KQ, and GC soils, respectively) due to its higher soil pH (9.0). There are three main reasons for the lower $pCO_2$ relative to the atmosphere in the XLHT leachates: (1) due to the high pH of this soil, bicarbonate ($HCO_3^-$) rather than dissolved $CO_2$ (more abundant in acidic conditions) was the predominant form in soil leachates; (2) the high flux of precipitation (1 L in 3-4 h per EPE) largely diluted dissolved $CO_2$ in soil solutions rather than allowed respired $CO_2$ to accumulate in solution; (3) (maybe less importantly) respiration rate was rather low in the XLHT soils due to its lower SOC content (5.1–15.6 mg C m$^{-2}$ h$^{-1}$ relative to 16–49 mg C m$^{-2}$ h$^{-1}$ in the KQ and GC soils; Fig. S3).

To clarify, the paragraph is revised:

"*Isotopic fractionation of leached DIC due to $CO_2$ loss in an open system is insignificant when the partial pressure of $CO_2$ ($pCO_2$) in the solution is lower than twice that of the surrounding atmosphere (Hendy, 1971; and Doctor et al., 2008). In the present study, $pCO_2$ in the XLHT leachates was low (~ 400 µatm assuming alkalinity equals to DIC concentration; Table S2) due to its high pH, low soil respiration and dilution of dissolved $CO_2$ under EPE, the influence of $CO_2$ outgassing on the $\delta^{13}C$ of leached DIC was thus considered not important.*"

**Response to Referee #2**

**General Comments:**

**Comment 1:**

The manuscript was significantly improved by the authors, they responded to all comments properly. There are two points, which should be more discussed, first: the transferability of the results to field conditions, especially the low water content in arid and semi-arid climates and what does it mean for soil carbon leaching.

**Response:**

Soil water content is indeed low in the temperate grasslands of Inner-Mongolia for the majority of the year. However, as we mentioned in the Materials and Methods, extreme precipitation events tend to occur in the summer with 70% of the annual precipitation falling within June-Aug (Fig. S2), which leads to oversaturation and leaching of soil carbon. There are normally 2-4 heavy precipitation events per summer (Fig. S2) in the study area, and our experiment design was consistent with the field precipitation frequency and strength. Furthermore, as we showed later, neither DOC nor DIC fluxes showed significant relationships with leachate volume during EPEs (Figs. 6e-f), indicating that we used sufficient amount of precipitation and these fluxes represent soil carbon's leaching potential under EPEs. These being clarified, we agree with the reviewer and add discussions on the implication of our findings for field conditions:

*"Interestingly, neither DOC nor DIC fluxes showed any significant relationships with the volume of leachates during EPEs (Figs. 6e-f). This indicates that we used sufficient amount of precipitation in this study to "scavenge" dissolved carbon from soils and hence these fluxes represent soil carbon's leaching potential under EPEs."*

*"Also, soil water content was set to be ~60% of max WHC initially in our experiment, higher than that in the field of temperate grasslands (XLHT and KQ). Thus, our measured DOC and DIC fluxes are likely to be higher than carbon leaching in the field due to greater water retention in drier soils. Hence, our estimate may represent an upper limit of soil carbon leaching potential under EPEs."*

**Comment 2:**

Second the comparison of soil respiration from bare vs. litter-covered soil.

**Response:**

Thank you for this suggestion. Comparison of soil respiration between litter-amended and non-amended soils is now added into Section 3.2:

*"Total respired $CO_2$ was higher in the litter-amended than non-amended soils before and after EPEs (Fig. S5). The cumulative respired $CO_2$ in the litter-amended XLHT, KQ, and GC soils were 16.7, 54.8, and 44.6 g C $m^{-2}$ during three EPEs, 20%, 22%, and 15% higher than that of the non-amended soils, respectively. Due to the wide presence of litter coverage in our studied soils, litter effect on soil respiration should be considered when estimating carbon budgets for these grassland soils."*

**Specific Comments:**

Additionally there are some points which are confusing, and for a final publication these parts should be

rewritten.

**Comment 3:**

Page 7, line 29: The authors present the soil respiration normalized for different SOC contents, with 2.2, 2.6 and 7.6 [µg C g⁻¹ SOC h⁻¹], it is not clear which values are presented, maximum or average values – please specify.

Furthermore, I think the specific respiration rate of the XLHT site is too high with 7.6 [µg C g⁻¹ SOC h⁻¹], given the maximum respiration rate of 13.7 [mg C m⁻² h⁻¹] (same page line 27) and SOC stocks of 7.4 [kg C m⁻²] (based on the numbers given in table 1). There must be a mistake in the calculation or in the presented numbers. In consequence, SOC of the XLHT site is not easier to degrade.

**Response:**

We reported maximum respiration rates herein and we indeed made a mistake in presenting the numbers. Specific soil respiration rates should be 2.2, 2.6, and 2.0 µg C g$^{-1}$ SOC h$^{-1}$ in the GC, KQ, and XLHT soils, respectively. Detailed parameters used for the calculation of specific respiration rate were shown in the following table:

| Station | Maximum soil respiration rate (mg C m$^{-2}$ h$^{-1}$) | SOC (g) | Bottom area of column (m$^2$) | Maximum specific respiration rate (µg C g⁻¹ SOC h⁻¹) |
|---|---|---|---|---|
| GC | 37.30 | 132 | 0.00785 | 2.21822 |
| KQ | 40.60 | 123 | 0.00785 | 2.59114 |
| XLHT | 13.70 | 55 | 0.00785 | 1.95536 |

Accordingly, we have revised this sentence as follows:

*"The maximum specific soil respiration rates normalized to SOC were 2.2, 2.6, and 2.0 µg C g$^{-1}$ SOC h$^{-1}$ in the non-amended GC, KQ, and XLHT soils, respectively. Therefore, SOC degradability was quite similar in the alpine and temperate grassland soils."*

**Comment 4:**

Page 8, line 3:

"Total respired $CO_2$ was higher in the litter-amended than non-amended soils before and after EPEs (Fig. S3), likely caused by one or two following reasons: (1) the degradation of labile components in the fresh litter; (2) increased degradation of SOC primed by litter additions (Fröberg et al., 2005; Ahmad et al., 2013). These two reasons might affect $CO_2$ release in litter-amended soils in two ways: (1) both of them had positive effects; (2) one of them had positive effects and the other one had negative ones, but the total net effects was positive."

The second sentence is confusing and makes no sense, I'd suggest to delete it and rewrite the first a little bit.

Suggestion: "… (1) the degradation of labile compounds in the fresh litter; (2) induced priming effects due to the addition of an easily available energy source."

**Response:**

Thank you. Revised accordingly.

**Comment 5:**

Page 8, line 3-18:

In that paragraph, I really miss the comparison between soil carbon loss from bare soil versus litter covered and the resulting discussion. What are the field conditions for the studied area is the soil normally covered by litter or not? What does it mean for SOC losses due to respiration and leaching? - As mentioned in the introduction the authors want to compare soil carbon loss from bare vs. litter-covered soil.

**Response:**

The following is added to this section:

*"Respired $CO_2$ was higher in the litter-amended than non-amended soils before and after EPEs (Figs. S3 and S5). The cumulative respired $CO_2$ in the litter-amended XLHT, KQ, and GC soils were 16.7, 54.8, and 44.6 g C $m^{-2}$ during three EPEs, 20%, 22%, and 15% higher than that of the non-amended soils, respectively. Due to the wide presence of litter coverage in our studied soils, litter effect on soil respiration should be considered when estimating carbon budgets for these grassland soils."*

**Comment 6:**

Page 8, line 25: "The specific EPE-induced CO2 was significantly different for the litter-amended soils among sites (p < 0.05), showing a pattern of KQ > XLHT > GC."

In my opinion you can not compare the specific EPE-induced $CO_2$ release of the litter amended soils, because you have no information about the source of $CO_2$ especially on the XLHT and KQ site. Due to the coarser soil texture of these two sites, it might be also possible that labile C is dissolved from the litter, transported down column and led to positive priming effects in the deeper parts of the column. However, since you have no information about the contribution of SOC mineralization to total $CO_2$ this statement is highly speculative. You can only compare the non-amended sites. It would be more interesting if you could show the reduction of SOC mineralization of the GC site, after EPE. Here you have the isotopic data.

**Response:**

This is a good point. As the respired $CO_2$ from litter-amended soils originated from both SOC and litter mineralization, it is not appropriate to normalize respiration to SOC. We have therefore deleted relevant results and discussion.

SOC-derived $CO_2$ flux in the litter-amended GC soils ranged from 6.2 to 15.7 mg C $m^{-2} h^{-1}$ after the first EPE, with specific SOC-derived $CO_2$ flux of 0.4-0.9 µg C $h^{-1}$ $g^{-1}$ SOC. By comparison, $CO_2$ flux in the non-amended GC soils ranged from 17.2 to 27.1 mg C $m^{-2} h^{-1}$, with specific $CO_2$ flux of 1.0-1.6 µg C $h^{-1}$ $g^{-1}$ SOC. The specific SOC-derived $CO_2$ flux was hence lower in the litter-amended than non-amended GC soils, indicating that litter addition induced a negative priming effect on SOC mineralization.

The above results and discussion are added:

*"Similarly, the specific $CO_2$ flux derived from SOC was lower in the litter-amended GC soils (0.4–0.9 µg C $h^{-1}$ $g^{-1}$ SOC) than in the non-amended GC soils (1.0–1.6 µg C $h^{-1}$ $g^{-1}$ SOC), further proving the negative priming effect."*

**Comment 7:**

Page 8, line 32:

"These results suggest that SOC contents and SOC:N ratios are important factors influencing the total EPE induced $CO_2$ release"

I disagree; figure 3b shows that there is no significant difference in specific respiration of the non-amended sites. Therefore, differences between sites might be less important.

**Response:**

We have removed this sentence.

**Comment 8:**

Page 9, line 2

For me it it not clear what the authors mean with " … a total of 0.57−0.71, 0.56−0.94, and 0.73−0.89 L of leachates were collected for the XLHT, KQ, and GC soils, respectively."

Does it mean e.g.

• that after 3 EPE in total 0.57 L of leachates were collected for the XLHT site?

• or does it mean a range, that after the first EPE 0.57 L were collected for the XLHT site and after the third EPE 0.71 L

Is there any explanation why the XLHT soil retain more water than the GC soil, due to the lower sand content in GC I would expect the highest retention on this site.

**Response:**

First, to clarify, there were 0.57, 0.71 and 0.69 L of leachates after the addition of 1 L of precipitation during the first, second, and third EPEs for the XLHT soils, respectively. Similarity, leachate volumes were 0.56, 0.94 and 0.83 L for the KQ soils and 0.73, 0.87 and 0.89 L for the GC soils, respectively. Such information is shown in Figure 4.

Second, although water holding capacity (WHC) is primarily controlled by soil texture, structure, and organic matter, water movement in soils is also affected by preferential flow (an uneven and rapid movement of water in soils; Soares et al., 2015). Higher clay contents tend to facilitate the formation of preferential flow in soils (Karup et al., 2016), which can also be triggered by heavy rainfall events (McGrath et al., 2009). The GC soils have a higher clay content than the other two, presumably causing potential preferential flows and thus a higher leaching volume.  In addition, although we added the same amount of water (10 mL) to the surface of all soil columns daily, it is difficult to ensure a constant water content for such large soil columns. Hence, water evaporation may have also differed for soils with different textures (low water loss from clay soils; Harris and Robinson, 1916) and the GC soils, with the highest clay and silt contents, needed less precipitation to compensate for the evaporated water loss. Regardless of the cause for varied leachate volumes from different soils, neither DOC nor DIC fluxes showed significant relationships with leachate volume during EPEs (Figs. 6e-f), indicating that we used sufficient amount of precipitation in our experiment and these fluxes represent soil carbon's leaching potential under EPEs.

To make this paragraph clear, it is now revised as follows:

*"During the first EPE, a total of 0.57, 0.56 and 0.73 L of leachates were collected from the XLHT, KQ, and GC soils, respectively. The leachate increased to 0.71, 0.94 and 0.87 L during the second EPE and was 0.69, 0.83 and 0.89 L during the third EPE, respectively (Fig. 4). Soil water content was set to be ~60% of max WHC before the first EPE, and leaching did not occur until soil water reached saturation. Therefore, the leachate volume was lowest during the first EPE and similar for the second and third EPEs. There were some variations in the volume of leachates from different soils, possibly related to preferential flows created during EPEs in the soil columns (McGrath et al., 2009) and water evaporation between EPEs."*

**Comment 9:**

Page 9, line 16:
"However, for the KQ soil having a relatively low SIC content which was similar to the added litter-OC (0.7 g per column; Table 1), the influence of litter addition on the DIC flux was quite obvious. Therefore, although the contribution of dissolved $CO_2$ to DIC fluxes should be more important in high-pH soils, the relative effect of litter amendment on DIC fluxes under EPEs seemed more significant in soils with a low SIC content."
I do not see the "quite obvious influence" of litter addition to DIC flux for the KQ site in figure 4b. Therefore I would recommend to give a number on the relative effect of litter amendment on DIC fluxes. Furthermore, comparing the GC with KQ site, the absolute / relative effect of litter addition on DIC fluxes was higher/equal on the GC site. Which contradicts the statement "the relative effect of litter amendment on DIC fluxes under EPEs seemed more significant in soils with a low SIC content."

**Response:**

To better clarify, this section is revised:

*"However, for the KQ soil having a relatively low SIC content similar to the added litter-OC (0.7 g per column; Table 1), litter amendment had a significant effect on the DIC flux (p < 0.05), increasing by 21 ± 13% and 15 ± 7% relative to the non-amended KQ soils during the second and third EPEs, respectively. There was also a 30 ± 19% increase in the DIC flux from the litter-amended GC soils relative to its non-amended counterpart during the second EPE. Therefore, litter amendment had a significant influence on DIC fluxes from soils with a relatively low SIC content (KQ and GC) under EPEs compared with the high-SIC XLHT soil."*

**Comment 10:**

Page 10, line 1:
"… despite their higher SOC contents and CO2 release rates (fig S5)."
Figure S5 does not show a CO2 release rate, See comment to Figure S5.

**Response:**

Sorry, there was a mistake—we referred to Figure S3. This is corrected and we have re-checked all other figures as well.

**Comment 11:**

Page 10, line 12 ff.:

"In addition, DIC loss exclusively resulting from SIC dissolution or weathering was also a significant fraction of soil carbon loss, equivalent to 219% SOC loss in the form of CO2 during EPEs (Fig. 5). These results collectively corroborate that inorganic carbon loss is the main form of soil carbon loss in alkaline soils during EPEs."

In my opinion this paragraph is a little bit confusing and needs more information especially the meaning of CO2 loss. I guess the authors mean here EPE induced CO2 release, however in the current form I would understand "SOC loss in form of CO2" as total respiration. I'd suggest to stay with the term "EPE induced CO2 loss".

But if the authors mean total CO2 release during EPE the presented numbers are wrong.

E.g. In figure 5 the CO2 loss during the first EPE was ca. 1 g C m⁻², however given the numbers from figure S3 for the first EPE for the XLHT site, basal respiration with ca. 7.8 mg C m⁻² h⁻¹. Calculating a CO2 loss for the first 20 days, would lead to 3.7 g C m⁻² for the site.

Furthermore, it is not clear which treatment is shown in figure 5 – non-amended or amended?

**Response:**

Thank you. We meant "EPE induced CO2 loss" and consistently use the term here.

The treatment shown in figure 5 was the non-amended XLHT soils.

To make this sentence and Figure 5 clearer, we have revised these parts as follows:

*"In addition, DIC loss exclusively resulting from SIC dissolution or weathering was also a significant fraction of soil carbon loss, equivalent to 219% SOC loss in the form of EPE-induced $CO_2$ during EPEs (Fig. 5). These results collectively corroborate that inorganic carbon loss is the main form of soil carbon loss in alkaline soils during EPEs."*

[Figure]

*Figure 5: Carbon loss fluxes from soil organic carbon (SOC) mineralization in the non-amended XLHT soils. Fluxes include extreme precipitation event (EPE)-induced $CO_2$ release and leaching of biogenic dissolved inorganic carbon (DIC), dissolved organic carbon (DOC) and lithogenic DIC. Mean values are shown with standard error (n = 3)."*

**Comment 12:**

Page 12, line 7:
Here I miss the discussion about the adjusted WHC of the soil columns, which might be important to estimate the losses of soil carbon in these arid and semi arid climates. In fact, under field conditions I'd expect a much lower DIC loss because under field conditions the water content in soils might be lower <

60% of WHC and soils can retain more water.

On page 9 line 2, the author mentioned that the sum of the collected leachates during 3 EPE was between 0.57 - 0.94 L, after the addition of in total 3 L precipitation. Therefore, this indicates a strong retention of rain water in the investigated soils, which of course depends on soil texture and water content before an EPE. It would be interesting to know how much water was added to adjust the WHC of the different soils. This would allow to give a more realistic estimation about soil carbon leaching losses under field conditions.

 Further, the time between two EPE might be also an important factor for DIC losses, because the longer the time between two EPEs the lower the soil water content. In consequence DIC losses might be smaller. Therefore, leaching of soil carbon might be smaller under field condition. In addition, after an EPE under field conditions the moisture content is altered and may provide better conditions for SOC mineralization which would increase CO2 losses.

**Response:**

Our original statements about leachate volume were a little confusing. In fact, there was 0.57 L of leachate after the addition of 1 L of precipitation during the first EPE for the XLHT soils. We have revised this paragraph completely to make it clearer—please see details in our reply to Comment 8.

As mentioned in our reply to Comment 1, soil water content is indeed very low in the field in the dry season but can be high in the summer due to the occurrence of heavy precipitation in this monsoon affected region. We designed the precipitation frequency and strength as close to the summer field condition as possible. However, the effect of low water content on DIC loss is important, especially in the dry season. Hence, we have added the following discussion to remind the readers of the limitations of our study:

 *"Also, soil water content was set to be ~60% of max WHC initially in our experiment, higher than that in the field of temperate grasslands (XLHT and KQ). Thus, our measured DOC and DIC fluxes are likely to be higher than carbon leaching in the field due to greater water retention in drier soils. Hence, our estimate may represent an upper limit of soil carbon leaching potential under EPEs."*

**Comment 13:**

Figure S4: Which bulk density was used for the GC site? There is no bulk density given in table 1. How did you calculated the numbers in figure S4.

**Response:**

Bulk density at the GC site was not determined. However, we measured the weight of wet soil in each soil depth. Combined with the field water content (FWC), soil inorganic carbon (SIC) and soil organic carbon (SOC) contents shown in Table 1, we can calculate soil carbon in each column. We have revised Figure S4 by adding the data of wet soil weight at each site.

[Figure]

*Figure S4: Total amount of soil organic carbon (SOC), soil inorganic carbon (SIC), and wet soil weight (WSW) in different layers of the artificial soil columns.*

**Comment 14:**

Figure S5: They y axis label and figure caption is wrong. The figure does not show a rate rather it show a sum of total CO2 produced during the three EPE.

**Response:**

We cited the wrong figure previously (see our reply to Comment 10; we referred to Figure S3 for rates). The axis label and figure caption in Figure S5 were actually consistent with our data, which showed the cumulative respiration during the three EPE.

[revised manuscript text omitted]

---

## Author Response (AR3)

Dear Editor,

Thank you for allowing us to submit a revised version of our manuscript. We greatly appreciate your detailed comments on improving the manuscript. We addressed all comments as attached below (highlighted in red for comments) and hope that we adequately solved the requests.

With kind regards,

Xiaojuan Feng

(We attached a marked version of the manuscript and the supplement with changes highlighted at the end of this pdf.)

**Response to Editor**

**Comment:**

P5,L24: please change "...using formula in Ran et al. (2015)." to "according to Ran et al. (2015)."

**Response:**

Revised.

**Comment:**

P6, L23: this sentence is unclear. I suggest the following change "In the present study, pCO2 in the XLHT leachates was low (~ 400 μatm assuming alkalinity equals to DIC concentration; Table S2) due to its high pH, low soil respiration and dilution of dissolved CO2 under EPE. Thus, we considered the influence of CO2 outgassing on the δ13C of leached DIC to be negligible."

**Response:**

Revised.

**Comment:**

P7, L21: please change to "......consistent with the "Birch Effect" (Birch, 1964),..."

**Response:**

Revised.

**Comment:**

P8, L14-15. This sentence seems overly obvious, because you divide CO2 flux data by the same soil C content for both the non amended and the litter-amended treatment. As such, I suggest deleting it.

**Response:**

Deleted.

**Comment:**

P10, L14: Please write CO2 with a lowercase "2"

**Response:**

Revised.

**Comment:**

P11,L6: No need to put the citations in italics.

**Response:**

Revised.

**Comment:**

P11,L26: please change "...used in our study has a.." to "used in our study have a "

**Response:**

Revised.

**Comment:**

P12,L12: Please write CO2 with a lowercase "2"

**Response:**

Revised.

**Comment:**

P12,L17: This sentence is twice repeated. Please delete.

**Response:**

Deleted.

**Comment:**

Fig.5: Please delete the double decimal on the right Y-axis.

**Response:**

Revised.

**Comment:**

Fig. S3: I strongly suggest turning the insert figure into a new, separate figure. This would improve readability. Also, the description of the figure would be much clearer that way (in your description of the lines, it's currently unclear when you are referring to the insert figure, or the other figures).

**Response:**

Thank you! The insert figure has been revised into a separate figure as Figure S4. All the citation of this regard in the manuscript has been revised accordingly.

**Comment:**

Fig. S3: what happened to the CO2 flux data for KQ, 2nd EPE? If no data were available for this EPE, then how did you produce Figure 3. Please explain.

**Response:**

Unfortunately, due to logistic reasons (the time for sampling overlapped with field trips), $CO_2$ release after

the second EPE in the KQ soils was not monitored. Instead, it was estimated as the average soil respiration after the first and third EPEs in the KQ soils. This is justified by the respiration data from the XLHT and GC soils, where the average respiration after the first and third EPEs was similar to the measured respiration after the second EPE (difference < 5% of total soil respiration for three EPEs; please see the following table).

| Soils | EPE | Cumulative respiration in non-amendment soils (mg C m$^{-2}$) | Cumulative respiration in litter-amendment soils (mg C m$^{-2}$) |
|---|---|---|---|
| XLHT | 1st EPE (measured) | 4821 | 6125 |
| | 2nd EPE (measured) | 5093 | 5956 |
| | 3rd EPE (measured) | 4040 | 4653 |
| | Total respiration during three EPEs | 13954 | 16734 |
| | 2nd EPE (estimated) [a] | 4431 | 5389 |
| | **Difference between the estimated and measured respiration relative to total respiration [b]** | **5%** | **3%** |
| KQ | 1st EPE (measured) | 14751 | 18010 |
| | 3rd EPE (measured) | 15152 | 18549 |
| | 2nd EPE (estimated) | 14952 | 18280 |
| GC | 1st EPE (measured) | 10550 | 11554 |
| | 2nd EPE (measured) | 12679 | 15589 |
| | 3rd EPE (measured) | 15470 | 17481 |
| | Total respiration during three EPEs | 38699 | 44624 |
| | 2nd EPE (estimated) [a] | 13010 | 14518 |
| | **Difference between the estimated and measured respiration relative to total respiration [b]** | **-1%** | **2%** |

[a]: the estimated respiration after the second EPE was calculated as the average respiration after the first and third EPEs.

[b]: Difference between the estimated and measured respiration relative to total respiration was calculated as (measured respiration – estimated respiration) / total respiration.

We have added relevant explanations in the Materials and Methods section as follows:

[revised manuscript text omitted]